# The Evaluation of Potential Anticancer Activity of Meloxicam—In Vitro Study on Amelanotic and Melanotic Melanoma

**DOI:** 10.3390/ijms26135985

**Published:** 2025-06-22

**Authors:** Marta Karkoszka-Stanowska, Zuzanna Rzepka, Dorota Wrześniok

**Affiliations:** Department of Pharmaceutical Chemistry, Faculty of Pharmaceutical Sciences in Sosnowiec, Medical, University of Silesia in Katowice, Jagiellońska 4, 41-200 Sosnowiec, Poland; zrzepka@sum.edu.pl

**Keywords:** meloxicam, amelanotic melanoma, melanotic melanoma, apoptosis

## Abstract

Meloxicam (MLX), a member of the non-steroidal anti-inflammatory drugs (NSAIDs), is a preferential inhibitor of cyclooxygenase-2 (COX-2) responsible for the synthesis of pro-inflammatory prostaglandins. MLX, due to its inhibition of the COX-2 enzyme, which is overexpressed in many cancers, including melanoma, leading to rapid growth, angiogenesis, and metastasis, represents a potentially important compound with anticancer activity. This study aimed to investigate the potential anticancer activity of meloxicam against amelanotic C32 and melanotic COLO 829 melanoma cell lines. The objective was achieved by assessing cell metabolic activity using the WST-1 assay and analyzing mitochondrial potential, levels of reduced thiols, annexin, and caspases 3/7, 8, and 9 by imaging cytometry, as well as assessing reactive oxygen species (ROS) levels using the H_2_DCFDA probe. The amelanotic melanoma C32 was more sensitive to MLX exposure, thus exhibiting antiproliferative effects, a disruption of redox homeostasis, a reduction in mitochondrial potential, and an induction of apoptosis. The results provide robust molecular evidence supporting the pharmacological effects of MLX, highlighting its potential as a valuable agent for in vivo melanoma treatment.

## 1. Introduction

Malignant melanoma is considered the most dangerous form of skin cancer due to its aggressive nature, high metastatic potential, and rapid spread to other organs, such as the lungs, liver, brain, or bones [1,2]. Melanoma typically originates from pigment-producing cells called melanocytes, whose main role is to protect the genetic material of skin cells from harmful external influences. In response to environmental stimuli—such as intense UV exposure, oncogenic mutations (e.g., BRAF), or biological agents—melanocytes can undergo clonal expansion and dedifferentiation, losing their characteristics and invading deeper layers of the skin [3,4,5]. Other factors predisposing one to skin cancer development include phenotype I or II, multiple pigmented nevi, geographical location, and immunosuppression [6]. The accumulation of pathogenic alterations in the genome and epigenome disrupts intracellular homeostasis and promotes melanoma development [7].

Despite growing global knowledge on melanoma pathogenesis, classification, and targeted therapy, epidemiological data remain unsatisfactory; melanoma ranks as the 19th most commonly reported cancer worldwide [8]. The highest incidence rates have been reported in Australia, New Zealand, Western Europe, and North America. If current trends continue, the global burden of melanoma is expected to rise by 50% in new cases and 68% in deaths by 2040, potentially making it a leading cause of cancer-related mortality [9,10]. Melanoma treatment has advanced significantly, with immunotherapies and targeted therapies contributing to improved survival rates. Surgery remains critical for early-stage melanoma, while advanced cases often require combination therapies [11]. For patients with BRAF V600 mutations, targeted therapies such as BRAF inhibitors (e.g., vemurafenib, dabrafenib) in combination with MEK inhibitors (e.g., trametinib, cobimetinib) have demonstrated improved progression-free and overall survival rates [12]. Immunotherapy, particularly immune checkpoint inhibitors like anti-PD-1 (nivolumab, pembrolizumab) and anti-CTLA-4 (ipilimumab), has revolutionized the treatment landscape by enhancing the host immune response against tumor cells [13]. Combination immunotherapy (e.g., nivolumab plus ipilimumab) has shown increased efficacy, albeit with higher toxicity profiles [14]. Additionally, adjuvant therapy with immune checkpoint inhibitors has been established as a standard for patients with resected stage III and IV melanoma [15]. Unfortunately, despite these advancements, challenges such as drug resistance and toxicity persist, driving the search for novel treatment strategies [11].

Cyclooxygenase-2 (COX-2) is an enzyme that plays a significant role in the inflammatory process and has been implicated in the progression of various cancers. COX-2 expression promotes angiogenesis, inflammation, and immune suppression. Additionally, it enhances the depth of invasion through mechanisms involving TGF-β1 and facilitates immune evasion by suppressing T-cell function by inducing myeloid-derived suppressor cells (MDSCs) [16]. In melanoma, COX-2 is a marker of tumor progression—elevated COX-2 expression is associated with increased tumor thickness, ulceration, and methastasis, leading to poorer survival rates among patients [17]. Given its role in tumor progression and immune suppression, COX-2 is a potential therapeutic target.

Meloxicam (MLX) is a selective COX-2 inhibitor, which belongs to the group of non-steroidal anti-inflammatory drugs (NSAIDs). The chemical structure of MLX is presented in Figure 1. The drug exhibits poor aqueous solubility, with a solubility of approximately 7.15 μg/mL in water at 25 °C, which presents challenges in formulation and bioavailability [18]. MLX is a weak acid with a pKa of approximately 1.1 (enolic group) and 4.2 (carboxamide group), contributing to its pH-dependent solubility profile [19]. The drug is lipophilic, exhibiting a logP value ca. 3.4, facilitating its membrane permeability and oral absorption. Meloxicam is highly bound to plasma proteins (~99%) and demonstrates a relatively long half-life of 15–20 h, allowing for once-daily dosing [20]. These physicochemical properties influence its pharmacokinetics, efficacy, and formulation strategies, including the development of solubility-enhancing delivery systems such as solid dispersions and nanocrystals.

MLX is widely used due to its analgesic and anti-inflammatory properties both in human and veterinary medicine for the treatment of skeletal systems. MLX is also used for the prevention of post-operative pain, especially in the form of fast-acting injections [21,22]. In addition to its primary analgesic activity, MLX demonstrates potential anticancer properties through COX-2 dependent and independent pathway. COX-2 dependent pathways of MLX anticancer activity involve prostaglandin E2 (PGE2) production inhibition which can be effective in cancers with high COX-2 expression such as osteosarcoma or hepatocellular carcinoma [23,24]. Invasion and migration of cancer cells may be reduced due to matrix metalloproteinase-2 (MMP-2) downregulation and E-cadherin upregulation by MLX. The presumed antineoplastic mechanisms of meloxicam, unrelated to effects on COX-2, include apoptosis induction by the upregulation of pro-apoptotic proteins, i.e., Bax, Fas-L, downregulation of anti-apoptotic proteins—Mcl-1 through inhibition of AKT phosphorylation. Additionally, MLX up-regulates Beclin-1 and IL-3 inducing autophagy and modulates the microRNA-200/PD-L1 pathway, increasing the sensitivity of tumor cells to immunotherapy [23,24,25].

To date, there are no published studies that have specifically examined the antitumor potential of meloxicam or its derivatives in the treatment of malignant melanoma. While meloxicam has demonstrated anticancer activity in other tumor types—such as non-small cell lung cancer and colorectal cancer—its effects on melanoma remain entirely unexplored in the scientific literature. This notable absence highlights a clear gap in current oncological research and underscores the novelty of the present study. By investigating meloxicam in the context of malignant melanoma, this research addresses an unmet need and explores a previously uncharted therapeutic avenue, potentially contributing valuable insights into melanoma treatment strategies. In view of the facts presented, the aim of the study was to evaluate the potential antitumor activity of meloxicam against amelanotic (C32) and melanotic (COLO 829) melanoma cells. Proliferative activity, reduced intracellular thiol content, mitochondrial potential, cell cycle, annexin, and caspase 3/7 were assessed in both melanoma lines tested. It was concluded that the amelanotic melanoma line C32 showed higher sensitivity to MLX, against which additional parameters—caspase 8 and 9, reactive oxygen species content and COX-2 expression—were examined.

## 2. Results

### 2.1. The Impact of Meloxicam on Proliferation of Melanoma Cells

In the first phase of the study, the impact of meloxicam on metabolic activity of melanoma cell population was examined by WST-1 probe. Tests were performed on cell types differentiated in their ability to synthesize the melanin biopolimer—amelanotic C32 and melanotic COLO 829. Prior to analysis, melanoma cells were incubated for 24, 48, and 72 h with meloxicam in wide concentration range—10–800 µM, selected on the basis of previously conducted studies on normal skin cells with varying degrees of pigmentation, as well as literature data using MLX in the treatment of other cancers in an analogous concentration range and time interval [26,27]. The results are presented in Figure 2. The study showed that meloxicam reduces the number of metabolically active cells in both tested cell lines. At the highest concentration tested, the number of metabolically active cells was reduced to approximately 20% regardless of the duration of exposure to MLX. The calculated EC50 values are, respectively, for C32 amelanotic melanoma 115 µM, 152 µM, and 112 µM for 24 h, 48 h, and 72 h. In the case of the melanotic melanoma COLO 829 EC50, for the different incubation times, the values are 24 h 270 µM, 48 h 232 µM, and 72 h 185 µM.

Analysis of cell counts showed the melantive activity of meloxicam proportional to the drug concentration used, as a significant reduction ca. 80% vs. control in cell counts was observed in both cases. Microscopic documentation of the experiment confirms the effects obtained from the WST-1 analysis—exposure to MLX caused a reduction in cell number, a change in cell shape to spherical, a loss of intercellular contacts and a tendency to detach.

### 2.2. The Assessment of Melanotic and Amelanoctic Melanoma Cell Cycle Exposed to Meloxicam

The assessment of the cell cycle in C32 and COLO 829 melanoma was carried out using the technique of image cytometry. The results presented in Figure 3 show that MLX induces changes in the cell cycle profile of both cell lines analyzed, C32 and COLO 829. MLX tended to reduce the percentage of cells in the G1/G0 phase in both melanotic and amelanotic melanoma at ca. 15–20% in concentrations of 500 and 700 µM. In addition, an increase in the percentage of cells in the G2/M phase was recorded for both melanotic melanoma by approximately 20% and amelanotic melanoma by ca. 12%. The obtained data suggests the occurrence of cell cycle arrest in G2/M phase and a reduced ability of cells to enter mitosis, as reflected in proliferation analysis using the WST-1 probe, cell counts, and microscopic images.

### 2.3. The Oxido-Reductive Status of Melanoma Cells Treated with Meloxicam

The primary function of glutathione (GSH) is to neutralize certain endogenous compounds, xenobiotics, and ROS. GSH, which is a proton donor for reactive and unstable molecules such as ROS, plays a key role in preventing damage to DNA and cellular proteins. Maintaining the correct balance between the oxidized and reduced forms of GSH is essential for the maintenance of intracellular oxido-reductive balance, the cell’s ability to survive under oxidative stress and normal cell function. The next step of the study involved analyzing the level of oxidized intracellular thiols in melanoma cells exposed to MLX which reflects the oxido-reductive homeostatic state of the cell using the image cytometry technique. The obtained results imaged in Figure 4A revealed that incubation with MLX increased oxidative transitions in amelanotic and melanotic melanoma cells; however, the observed effect was stronger in amelanotic cell line. In the C32 population incubated with MLX at a concentration of 700 µM, more than 65% of the cell population had oxidized reduced thiols. For melanotic melanoma cells exposed to identical conditions, the noted percentage was approx. 45%.

As greater disruption of intracellular oxido-reductive homeostasis has been reported for amelanotic melanoma, the content of oxygen free radicals (ROS) in this cell line was sequentially investigated using the fluorimetric spectrometry technique and the H_2_DCFDA probe (Figure 4C). MLX was found to induce an increase in ROS levels in proportion to the concentration used. At the highest concentration analyzed, 700 µM, this was approximately 520% relative to the control.

### 2.4. Analysis of Mitochondrial Membrane Potential in Melanoma Cells Exposed to Meloxicam

The mitochondrial membrane potential (ΔΨm), which results from oxido-reductive transformations associated with the Krebs cycle, is an intermediate form of energy storage used for ATP synthesis. Maintaining a stable level of mitochondrial potential and intracellular ATP is essential for proper cell function. Transient changes in the parameters described may be observed as a result of physiological activity of the cell, but long-term disturbances of both factors may lead to the development of pathological changes. Mitochondrial membrane potential was assessed using imaging cytometry techniques and the fluorescence probe JC-1, which allows differentiation between cells with high and low ΔΨm, is therefore a valuable indicator of reversible steps to apoptosis. As depicted in Figure 5A, meloxicam induced ΔΨm depolarization in melanoma cells. The observed effect was proportional to the drug concentration used. At a concentration of 700 µM, the drug increased the percentage of cells with depolarized mitochondrial membranes to approximately 47% in amelanotic C32 melanoma and about 33% in COLO 829 cells.

### 2.5. Assessment of Meloxicam-Induced Apoptosis in Melanoma Cells

The Annexin V assay is a credible method for detecting apoptotic cells. The characteristic feature of apoptotic cells is phosphatidylserine externalization, which forms the basis for the Annexin V binding assay. Quantitative assessment of apoptosis in melanoma cells incubated with MLX was performed involving Annexin V staining. Conducted analysis showed that MLX induced apoptosis in both types of melanoma, although the obtained values are higher for amelanotic melanoma. The results presented in Figure 6A show that MLX caused an almost 60% increase in the percentage of apoptotic C32 cells relative to the control at a concentration of 700 µM. In the case of melanotic melanoma, the observed difference is approx. 20% at the highest tested concentration.

### 2.6. Analysis of Caspases’ Activity in C32 and COLO 829 Melanoma Cells Treated with Meloxicam

The study of caspase activity was performed using fluorescent image cytometry. The technique utilizes fluorochrome-labeled caspase inhibitors that specifically attach to the enzymes, marking the analyzed cells. The examination revealed that MLX was able to induce activation of caspases (Figure 7). Amelanotic melanoma once again proved to be more sensitive for MLX treatment. The percentage of C32 cells with activated caspase 3/7 was ca. 70% and melanotic melanoma COLO 829 was approx. 25%. Due to the high activity of caspase 3/7 in amelanotic melanoma, the activities of initiator caspases 8 and 9 were assessed sequentially. The increase in caspase 8 activity after incubation with MLX at a dose of 700 µM was found to be around 40%, while caspase 9 showed an almost 80% increase in the percentage of cells with activated caspase.

### 2.7. The Evaluation of COX-2 Enzyme Expression in C32 Amelanotic Melanoma Cells

COX-2 enzyme expression was examined using confocal microscopy techniques. Analyses based on immunofluorescence imaging, in addition to the visualization of COX-2 protein in the cells tested, provided data indicating an increase in COX-2 expression in cells exposed to MLX at a dose of 700 µM compared to control cells (Figure 8).

## 3. Discussion

Malignant melanoma is considered the third most common form of skin cancer [28]. Although significant progress has been made in melanoma treatment, many patients with metastatic melanoma still face a poor prognosis and a high mortality rate—approximately 75%. The aggressive nature of this cancer, drug resistance, and side effects of available therapies highlight the urgent need for more effective anti-melanoma medical treatment [29]. New drug development is fraught with numerous challenges that can hinder the process and increase costs. These challenges include high research costs, low success rates, regulatory hurdles, and market competition. Drug repositioning is a cost-effective and time-saving strategy compared to traditional drug discovery that involves finding new therapeutic uses for existing drugs [30].

Multiple studies have demonstrated increased COX-2 levels in various human cancers, including melanoma. Recent findings suggest that COX-2 is overexpressed in malignant melanoma and may contribute to disease progression [28,31]. Meloxicam is a selective COX-2 inhibitor, which was used in the present study to determine the effect of the drug on malignant melanoma homeostasis.

In the initial phase of the study, the cytotoxic potential of MLX against melanoma cells was assessed. The results indicated that meloxicam reduced cell viability and inhibited cell proliferation in a concentration-dependent manner. The EC50 values for amelanotic C32 melanoma were 115 μM, 152 μM, and 112 μM for 24 h, 48 h, and 72 h of incubation, respectively. For melanotic melanoma COLO 829 EC50 values were 270 μM, 232 μM, and 185 μM for 24 h, 48 h, and 72 h, respectively. Comparing the data obtained from the present study with the estimated coefficients of EC50 values obtained when normal skin cells—melanocytes and fibroblasts—were exposed to MLX, it can be concluded that for normal cells the EC50 value is above 500 µM and above 1000 µM, respectively [26]. These findings suggest that MLX exhibits selective cytotoxicity toward tumor cells. Notably, the EC50 values obtained are comparable to, or even lower than, those of other drugs tested in melanoma cell models, e.g., fluoroquinolones: lomefloxacin (EC50_24h_ 510 μM, EC50_48h_ 330 μM, EC50_72h_ 250 μM), moxifloxacin (EC50_24h_ 400 μM, EC50_48h_ 220 μM, EC50_72h_ 150 μM), ciprofloxacin (EC50_24h_ 740 μM, EC50_48h_ 170 μM, EC50_72h_ 100 μM), or tetracyclines: doxycycline (EC50_24h_ 74 μM, EC50_48h_ 32 μM, EC50_72h_ 16 μM) [29,32,33,34]. The results obtained in the MLX study against the mouse melanoma cell line B16F10 indicate that the EC50 values obtained in the MTT mitochondrial survival assay are significantly higher than those obtained on the human melanoma lines C32 and COLO 829, being, respectively, EC50_24h_ 760 μM, EC50_48h_ 590 μM, and EC50_72h_ 650 μM [35]. In in vitro studies, meloxicam is often used at higher concentrations than those typically administered in vivo. This adjustment is necessary due to the specific nature of in vitro conditions, where higher drug doses are required to produce observable and measurable effects on isolated cells or tissues. Unlike in vivo systems, where pharmacokinetics and systemic metabolism influence drug availability, in vitro environments lack these dynamic processes, potentially reducing the apparent potency of the compound. Therefore, to achieve relevant biological responses and investigate the mechanisms of action effectively, elevated doses of meloxicam are applied. However, findings from such studies must be carefully interpreted, as translating these results to in vivo settings in animals or humans will require dose modifications to account for physiological and safety considerations.

Reactive oxygen species are highly bioactive molecules that have been extensively studied in the aspect of carcinogenesis. These molecules are natural byproducts arising from many intracellular processes [36]. Cancer cells exhibit elevated ROS levels compared to normal cells due to the imbalance between pro- and antioxidants. In the cellular metabolism ROS are signaling molecules that promote cell proliferation, migration, invasion, and angiogenesis [37,38,39]. However, excessive ROS accumulation can lead to cellular damage by targeting proteins, nucleic acids, lipids, membranes, and organelles, ultimately resulting in cell death. Elevated ROS levels can induce cell cycle arrest, thereby triggering apoptosis [40]. To counteract elevated intracellular ROS, cancer cells utilize both low molecular weight scavengers such as glutathione (GSH), and specific antioxidant enzymes—superoxide dismutase, glutathione peroxidase, and catalase—whose transcription is upregulated in response to high ROS levels [41]. Glutathione, a widely occurring thiol, serves as a vital intracellular antioxidant and helps maintain cellular redox balance. In its reduced state, GSH defends cells against oxidative stress caused by reactive oxygen and harmful xenobiotics. Additionally, glutathione plays a role in regulating key cellular processes, including signaling pathways, cell growth, differentiation, and programmed cell death [42,43].

The assessment of the oxidized level of reduced thiols demonstrated that MLX significantly increased melanoma cell number with oxidized reduced thiols. It was found that amelanotic melanoma C32 was more potent than melanotic cell line COLO 829. Approximately 70% of C32 cells and 50% of COLO 829 cells treated with 700 µM MLX showed high levels of oxidized thiols. In addition, an analysis of the ROS content showed that an approximately five-fold increase in ROS content was observed in the more MLX-susceptible amelanotic melanoma line compared to control samples. A broad analysis of the results suggests that the reduction in thiol levels and ROS content correlate with the proliferation and cell count of the melanoma. By contrasting the results obtained with the analysis of the effect of MLX on the oxido-redox homeostasis of normal skin cells—melanocytes and fibroblasts, it can be concluded that the drug generates a much greater disturbance of redox homeostasis in melanoma cells. The level of oxidized thiols in melanocytes exposed to MLX at a dose of 1000 µM was about 20%, while in fibroblasts it was about 26%. Regarding ROS content, MLX at 1000 µM caused an approximately 50% increase in free radical density in fibroblasts and about 80% in melanocytes [26]. These findings indicate that MLX has a favorable safety profile for normal skin cells, inducing more pronounced effects in melanoma cells even at lower doses. Similar results have been reported with other NSAIDs, such as diclofenac and piroxicam, which increased ROS levels in the SK-MEL-5R melanoma cell line showing antitumor potential [44]. The ability to manipulate ROS levels demonstrates considerable therapeutic potential against cancer.

A commonly accepted view is that the reduction in intracellular GSH levels is a hallmark characteristic of cells undergoing apoptosis. Mitochondria are crucial organelles in triggering apoptosis, acting both as a source and a target of reactive oxygen species. Apoptosis, or programmed cell death, is a key target in cancer therapy and can be triggered via intrinsic or extrinsic pathways. The intrinsic, mitochondria-dependent pathway involves cytochrome c release, which forms the apoptosome with Apaf-1 and procaspase-9, leading to activation of caspase-9 and effector caspases 3 and 7. The extrinsic pathway is initiated by death receptors (e.g., TNF family), which activate caspase-8 and subsequently effector caspases. Regardless of the pathway, apoptosis results in characteristic features such as chromatin condensation, cell shrinkage, membrane blebbing, loss of adhesion, and phosphatidylserine exposure. In the present study, the assessment of phosphatydyloserine externalization by annexin V assay was carried out to confirm MLX-induced apotosis in melanoma cells. MLX significantly increased the proportion of annexin V-positive cells, with a higher percentage of apoptotic cells observed in amelanotic melanoma at 700 µM.

The perturbation of the transmembrane mitochondrial potential indicates that the apoptosis process is induced in an intrinsic pathway. The drug in a concentration of 700 µM increased the percentage of cells with depolarized mitochondrial membranes up to approx. 47% in amelanotic C32 melanoma and ca. 33% in the case of COLO 829 cell line. A self-reinforcing cycle between mitochondrial membrane depolarization and the buildup of reactive oxygen species within the mitochondria was demonstrated by Suzuki-Karasaki et al. [45]. The researchers found that an increase in mitochondrial ROS can intensify membrane depolarization, while agents that disrupt the mitochondrial membrane potential also lead to elevated ROS levels. This suggests that MLX’s effects—namely its promotion of oxidative stress and disruption of mitochondrial membrane potential—may be closely linked through this feedback mechanism.

The clear evidence of apoptosis in amelanotic melanoma cells following exposure to MLX confirmed by caspase 3/7 activation has prompted continued research into determining the pathway of induction of this process. To achieve this, the activities of caspase-8 and caspase-9 were assessed. The results of the assay indicated that MLX is capable of triggering both proapoptotic pathways in amelanotic melanoma cells. Once again, amelanotic melanoma cells appeared to be more sensitive to the apoptotic effects of MLX and the effect increased in relation to the concentration of the drug. Other COX-2 inhibitors, such as celecoxib, can reduce melanoma cell viability, proliferation, and metastatic potential in preclinical models [46]. The anti-melanoma effect of NSAIDs is thought to be mediated not only through COX-2 inhibition but also via COX-independent pathways, including modulation of mitochondrial apoptosis and suppression of Akt signaling [47]. Additionally, NSAIDs may enhance the effectiveness of existing therapies, including immune checkpoint inhibitors, by reducing prostaglandin-mediated immunosuppression in the tumor microenvironment [48]. Studies have shown that other NSAIDs, including ibuprofen, mefenamic acid, naproxen or diclofenac, which are commonly used for their anti-inflammatory and analgesic properties, are capable of modulating apoptosis by affecting annexin V and caspases 3/7 and 9 in cholangiocarcinoma and liver cancer cells [49,50,51].

Inhibition of COX-2 is molecularly associated with the induction of apoptosis and the disruption of redox homeostasis through several interconnected pathways. COX-2 catalyzes the conversion of arachidonic acid to prostaglandins, particularly prostaglandin E2 (PGE2), which promotes cell survival by activating pro-proliferative and anti-apoptotic signaling cascades, including the PI3K/Akt and ERK1/2 MAPK pathways [52]. Suppression of COX-2 leads to decreased PGE2 synthesis, resulting in downregulation of Bcl-2 family anti-apoptotic proteins (e.g., Bcl-2, Bcl-xL) and upregulation of pro-apoptotic factors such as Bax and caspase-3 activation, thereby initiating the intrinsic (mitochondrial) apoptotic pathway [53]. Moreover, COX-2 inhibition significantly impacts the redox state of the cell. Under physiological conditions, COX-2 modulates ROS levels and affects the expression of antioxidant enzymes such as superoxide dismutase (SOD) and glutathione peroxidase (GPx) [54]. Inhibition of COX-2 disrupts this balance, often resulting in excessive accumulation of reactive oxygen species (ROS), which damages lipids, proteins, and DNA, and promotes mitochondrial membrane depolarization, cytochrome c release, and subsequent activation of caspase-dependent apoptosis [55]. Additionally, elevated ROS levels can modulate redox-sensitive transcription factors, including NF-κB and AP-1, which control the expression of genes related to inflammation, survival, and apoptosis. COX-2 inhibition may attenuate NF-κB activation, leading to reduced transcription of survival genes and a shift toward apoptotic cell fate [56]. Thus, COX-2 inhibitors such as meloxicam not only suppress inflammatory signaling but also promote apoptosis and oxidative stress, contributing to altered redox homeostasis and cell death in both pathological and therapeutic contexts.

In the present analysis, the inductive effect of MLX on COX-2 expression was demonstrated. In melanoma cells exposed to the COX-2 inhibitor meloxicam, a paradoxical increase in COX-2 expression _can_ occur via several mechanisms. First, COX-2 inhibition leads to reduced prostaglandin E_2_ (PGE_2_) levels, prompting a compensatory upregulation of COX-2 gene transcription as the cell attempts to restore homeostasis. Additionally, meloxicam induces oxidative and inflammatory stress, activating transcription factors such as NF-κB, MAPKs, and AP-1, which are well-established regulators of COX-2 expression. This adaptive response is further reinforced by COX-independent pathways: meloxicam has been shown to modulate autophagy and influence signaling molecules such as STAT3, SIRT1, and AMPK, all of which can feedback to enhance COX-2 transcription [57]. In some cancer models, partial inhibition of COX-2 by meloxicam fails to suppress PGE_2_ fully, maintaining a proliferative signaling loop that sustains COX-2 expression [24]. Thus, the observed COX-2 upregulation reflects a multifaceted adaptive and compensatory cellular response involving both COX-dependent and independent regulatory networks.

Notably, melanotic and amelanotic melanoma cell lines differ significantly in their pigmentation levels, reflecting cellular differentiation and different molecular signatures. These differences may have a critical impact on the cellular response to both targeted therapies and COX-2 inhibitors. Melanotic melanoma cells often retain functional melanogenesis pathways and carry mutations commonly found in cutaneous melanoma, such as BRAF V600E, NRAS or NF1 [47,48]. These mutations promote sustained activation of the MAPK signaling cascade, often resulting in elevated COX-2 expression [58]. Consequently, melanotic lines tend to exhibit higher baseline levels of COX-2 and are generally more susceptible to COX-2 inhibition, which may inhibit proliferation and inflammation-related survival pathways [59]. In contrast, amelanotic melanoma cells tend to be less differentiated and are often characterized by reduced expression of pigmentation-related genes (e.g., MITF, TYR) and alternative oncogenic alterations. These may include abnormal activation of the PI3K/AKT pathway or loss-of-function mutations in tumor suppressors such as TP53, potentially leading to reduced COX-2 expression and reduced sensitivity to its inhibition [47]. Nevertheless, some amelanotic cell lines may exhibit compensatory up-regulation of inflammatory mediators, making them partially sensitive to a COX-2 blockade, especially when combined with other targeted or immunomodulatory agents [59]. In summary, mutation diversity and differentiation status of melanoma cells play a key role in modulating COX-2 expression and activity, which in turn influences the therapeutic efficacy of COX-2 inhibitors. These insights support a precision oncology approach to selectively use COX-2-targeting strategies based on tumor subtype and molecular profile.

MLX has shown promising anticancer activity in multiple cancer types. Its mechanism of action involves both COX-2-dependent and COX-2-independent pathways, influencing processes such as cell growth, programmed cell death, and tumor development. MLX suppresses the migration, invasion, and colony-forming ability of hepatocellular carcinoma cells by regulating the expression of E-cadherin and MMP-2. It promotes apoptosis by increasing pro-apoptotic protein levels while decreasing anti-apoptotic protein expression, engaging both COX-2-dependent and independent mechanisms. Furthermore, under hypoxic conditions, meloxicam disrupts mitochondrial membrane potential and induces cell death [25]. Moreover, the anticancer potential of MLX has been demonstrated against the tumors, i.e., colorectal cancer, osteosarcoma, non-small cell lung cancer, and Burhitt Lymphoma leading to inhibition of growth, invasiveness, and proliferation of cells [60,61,62,63].

## 4. Materials and Methods

### 4.1. Chemicals

Meloxicam was obtained from Boehringer Ingelheim (purity ≥ 98% (HPLC)) (Budapest, Hungary). Dulbecco’s Modified EagleMedium (DMEM), Roswell Park Memorial Institute (RPMI) 1640 medium, Trypsin/EDTA (0.25%/0.02%), and Fetal Bovine Serum (FBS) were obtained from PAN-Biotech GmbH (Aidenbach, Germany). H_2_DCFDA reagent, Dulbecco’s phosphate-buffered saline (DPBS), Penicillin-Streptomycin solution (10,000 U/mL) (purity: ≥97–≥99% (HPLC)), anti-rabbit secondary antibody Alexa Fluor 488 conjugate were purchased from Thermo Fisher Scientific Inc. (Waltham, MA, USA). WST-1 cell proliferation reagent was obtained from Roche GmbH (Mannheim, Germany). Solution 3 (DAPI 1 µg/mL, triton X-100 0.1%), Solution 5 (VitaBright-48 400 µg/mL, propidium iodide 500 µg/mL, acridine orange 1.2 µg/mL), Solution 7 (JC-1 dye 200 µg/mL), Solution 8 (DAPI 1 µg/mL), Solution 15 (Hoechst 33342 500 µg/mL), Solution 16 (propidium iodide 500 µg/mL), Via1-Cassettes, and NC-Slides A2 and A8 were purchased from ChemoMetec (Lillerød, Denmark). Annexin V-CF488A was obtained from Biotium (Fremont, CA USA). 10x Annexin V binding buffer was obtained from BioVision Inc. (Milpitas, CA, USA). Green Fluorescent FAM-FLICA Caspase-3/7 Assay Kit, Green Fluorescent FAM-FLICA Caspase 8 Assay Kit, and Green Fluorescent FAM-FLICA Caspase 9 Assay Kit were obtained from ImmunoChemistry Technologies (Bloomington, MN, USA). COX2 (D5H5) XP Rabbit mAb was obtained from Cell Signaling Technology (Danvers, MA, USA). Bovine Serum Albumin (BSA), Phalloidin-Atto565 were purchased from Sigma Aldrich (St. Louis, MO, USA). The fluorescence mounting medium was obtained from Dako Denmark A/S (Glostrup, Denmark).

### 4.2. Cell Culture

In this study, human melanoma cell lines COLO 829 and C32 were utilized. Both cell lines were obtained from ATCC (Manassas, VA, USA) and maintained in culture at 37 °C in a 5% CO_2_ atmosphere. COLO 829 melanotic melanoma cells were grown in RPMI medium with L-glutamine supplemented with 10% fetal bovine serum (FBS), while C32 amelanotic melanoma cells were cultured in DMEM supplemented with 10% FBS. To prevent contamination 100X penicillin-streptomycin solution was added to the culture media (final concentrations: 100 units/mL of penicillin and 100 μg/mL of streptomycin).

### 4.3. WST-1 Assay

Cytotoxicity of meloxicam on melanoma cells was assessed using the Cell Proliferation Reagent I (WST-1). Cells were seeded in 96-well plates (2.5 × 10^3^ cells per well) and incubated in culture medium for 24 h. After incubation, the culture medium was removed, and meloxicam solutions in medium (100 µL per well) were added for 24 h, 48 h, or 72 h. Three hours before measurement, WST-1 reagent (10 µL per well) was added into each well. Absorbance was recorded at 440 nm and 650 nm using an Infinite 200 PRO microplate reader (TECAN, Männedorf, Switzerland). This experiment was performed in three independent repetitions. The results were expressed as a percentage relative to the control.

### 4.4. Cell Count and Viability Assay

The NucleoCounter NC-3000 image fluorescence cytometer (ChemoMetec, Lillerød, Denmark) was used to evaluate the cell count and viability of C32 and COLO 829 melanoma cells exposed to meloxicam (300, 500, 700 µM for 48 h). In summary, cell cultures were detached using trypsinization, centrifuged, and then resuspended in a growth medium. The prepared cell suspensions were loaded into Via1-Cassettes filled with two fluorescent dyes: acridine orange, which labels all cells, and DAPI, which selectively stains dead cells. The analysis was performed following the “Cell Viability and Cell Count Assay” protocol.

### 4.5. Cell Cycle Assessment

The cell cycle was examined using the NucleoCounter^®^ NC-3000™ fluorescence imaging cytometer (ChemoMetec, Lillerød, Denmark), which enables assessment based on intracellular DNA content. Measurements were taken after 48 h treatment with MLX. A total of 1 × 10⁶ melanoma cells were suspended in 0.5 mL of PBS, then fixed by adding 4.5 mL of 70% ethanol, with incubation at 0–4 °C for 12 h. After fixation, the cells were centrifuged, the resulting pellets resuspended in PBS, and centrifuged again for 5 min at 500× *g*. The cells were subsequently stained with Solution 3, following the manufacturer’s instructions. Stained samples were loaded onto NC-Slides A8™ and analyzed using the Fixed Cell Cycle-DAPI Assay protocol provided by the NC-3000 system. The resulting histograms allowed for identification of different cell cycle phases.

### 4.6. Analysis of the Intracellular Thiol Status

The intracellular thiol levels were assessed by VitaBright-48 staining. This reagent enables the identification of cell populations with high levels of reduced thiols, of which GSH is the main one. In brief, 48 h after treatment with meloxicam, the melanoma cells were resuspended in PBS and stained with Solution 5 containing VitaBright-48, according to the manufacturer’s manual. The analysis was performed using the NucleoCounter NC-3000 image cytometer and NucleoView NC-3000 Software 2.1.25.12 (ChemoMetec, Lillerød, Denmark).

### 4.7. Analysis of Mitochondrial Membrane Potential

The mitochondrial membrane potential was assessed using JC-1 staining followed by image cytometry analysis. The fluorescent cationic dye JC-1 accumulates in mitochondria with high transmembrane potential. At elevated concentrations, JC-1 forms aggregates that emit red fluorescence, whereas in cells with low mitochondrial potential, the dye remains in the cytoplasm and exhibits green fluorescence. Briefly, after treatment with meloxicam, the melanoma cells suspensions were stained with Solution 7 containing JC-1, according to the manufacturer’s instruction. Just before analysis, the cells were resuspended in Solution 8 containing DAPI. The final analysis was performed using the NucleoCounter NC-3000 image cytometer and the NucleoView NC-3000 Software.

### 4.8. Intracellular Reactive Oxygen Species Detection

The intracellular reactive oxygen species level was assessed using the H_2_DCFDA indicator. This assay involves staining cells with 2′,7′-dichlorodihydrofluorescein diacetate (H_2_DCFDA). Inside the cells, H_2_DCFDA is deacetylated by esterases and then oxidized by ROS to 2′,7′-dichlorofluorescein (DCF), a compound that emits green fluorescence. Melanoma cells were seeded in 96-well plates and incubated in culture medium for 24 h. After incubation, the culture medium was removed, and meloxicam solutions in medium were added for 48 h. The cells were then incubated with H_2_DCFDA for 30 min, washed twice with PBS, and finally, the fluorescence intensity was measured using the Infinite 200 Pro microplate reader. The results were normalized to the number of metabolically active cells.

### 4.9. Anexin V Assay

The method used is based on the strong affinity of annexin V for phosphatidylserine, which is exposed on the outer membrane of apoptotic cells. Following treatment with meloxicam, melanoma cells were resuspended in Annexin V Binding Buffer (ABB) supplemented with Hoechst 33342 (Solution 15) and FITC-labeled annexin V (Annexin V-CF488A). The reagent volumes used were in accordance with the cytometer manufacturer’s manual “Annexin V Assay”. The cells were then incubated for 15 min at 37 °C and centrifuged. Afterward, the resulting cell pellets were washed twice with ABB. Finally, the pellets were resuspended in ABB supplemented with propidium iodide (Solution 16) to stain late apoptotic and necrotic cells. The analysis was performed using the NucleoCounter NC-3000 fluorescence image cytometer.

### 4.10. Analysis of Caspases Activity

Caspase 3/7, 8, and 9 activity was assessed cytometrically using Fluorescent Labeled Inhibitors of Caspases (FLICA) that covalently bind to active caspase enzymes. After treatment with meloxicam, melanoma cells were resuspended in PBS, FLICA reagent and Hoechst 33342 (Solution 15) were added, and then the mixture was incubated for 60 min at 37 °C, according to the ChemoMetec’s manual “Caspase Assay”. Following incubation, the samples were washed twice with Apoptosis Wash Buffer, centrifuged, and then resuspended in Apoptosis Wash Buffer supplemented with propidium iodide (Solution 16). Measurements were taken using a NucleoCounter NC-3000 cytometer controlled by NucleoView NC-3000 Software.

### 4.11. Immunocytochemistry and Confocal Imaging

C32 cells grown on sterile coverslips placed in Petri dishes were fixed with 4% paraformaldehyde and permeabilized with 0.1% Triton X-100. After blocking with glycine and 3% BSA solutions, cells were incubated with primary antibody (anti-COX2, dilution 1:400) overnight at 4 °C. Then the cells were incubated with Phalloidin–Atto 565 and the secondary antibody conjugated with Alexa Fluor 488 (dilution 1:200) for 2 h. The coverslips were mounted onto a microscopic glass slide. The samples were scanned using a Nikon A1R Si confocal imaging system with a Nikon Eclipse Ti-E inverted microscope.

### 4.12. Statistical Analysis

Statistical analysis was performed using GraphPad Prism 8.0 (Graph-Pad Software, San Diego, CA, USA). The Shapiro–Wilk test was used to evaluate normality of the data. Homogeneity of variances was verified by the Brown–Forsythe test. Statistical significance of differences between groups was evaluated using one-way ANOVA and Dunnett’s test. Statistical significance was determined by a *p*-value less than 0.05.

## 5. Conclusions

In conclusion, MLX demonstrates significant potential as an adjunct in anticancer therapy due to its multifaceted mechanisms of action. By inhibiting tumor cell proliferation, migration, and invasion, and promoting apoptosis through both COX-2-dependent and independent pathways, meloxicam offers a promising approach to targeting cancer progression. Its ability to enhance immunotherapy responses and disrupt hypoxia-related survival pathways further underscores its therapeutic value. However, the present research is limited by a scarcity of clinical trials, particularly in human patients, and a reliance on in vitro and animal models, which may not fully replicate the complexity of human malignancies. Moreover, the precise molecular mechanisms underlying meloxicam’s antitumor effects remain incompletely understood. Future research should focus on well-designed clinical studies to assess the efficacy and safety of meloxicam in cancer patients, as well as investigations into its mechanisms of action, optimal dosing strategies, and potential synergistic effects with other anticancer agents. Expanding research into its use in veterinary oncology, especially in canine cancers such as melanoma, may also offer valuable insights and translational relevance.

## Figures and Tables

**Figure 1 ijms-26-05985-f001:**
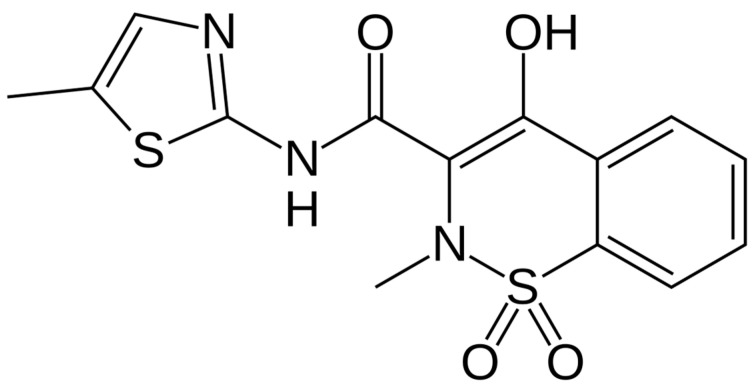
Chemical structure of meloxicam.

**Figure 2 ijms-26-05985-f002:**
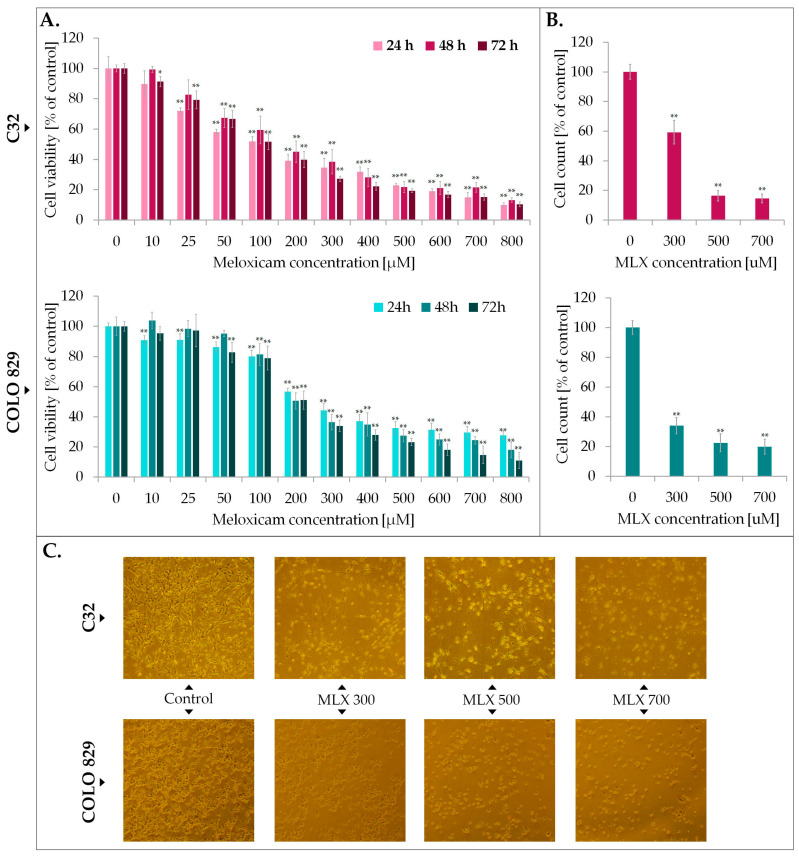
The impact of meloxicam (MLX) on proliferation (**A**), cell number (**B**), and morphology (**C**) of amelanotic (C32) and melanotic (COLO 829) melanoma. The analysis of cell viability was performed after 24, 48, and 72 h incubation of the cells with MLX at a concentration range of 10—800 µM. Cell count and morphology assessment were examined after 48 h of treatment with MLX at a concentration 300, 500, and 700 µM. Bar graphs show the mean value ± SD of three independent experiments; * *p* < 0.05; ** *p* < 0.01. In microscopic images scale bar = 100 µM.

**Figure 3 ijms-26-05985-f003:**
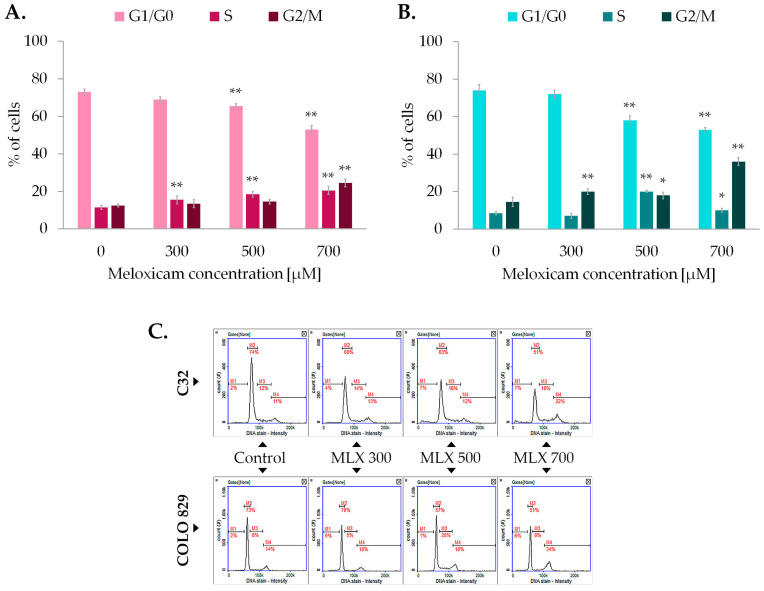
The impact of meloxicam (MLX) on the amelanotic (C32) (**A**) and melanotic (COLO 829) (**B**) melanoma cell cycle. All analyses were performed after a 48 h incubation of the cells with MLX at a concentration of 300, 500, or 700 µM. Representative histograms from the analysis (**C**) depict the distribution of the percentage of cells in the different phases of the cell cycle. Bar graphs show the mean value ± SD of three independent experiments; * *p* < 0.05; ** *p* < 0.01.

**Figure 4 ijms-26-05985-f004:**
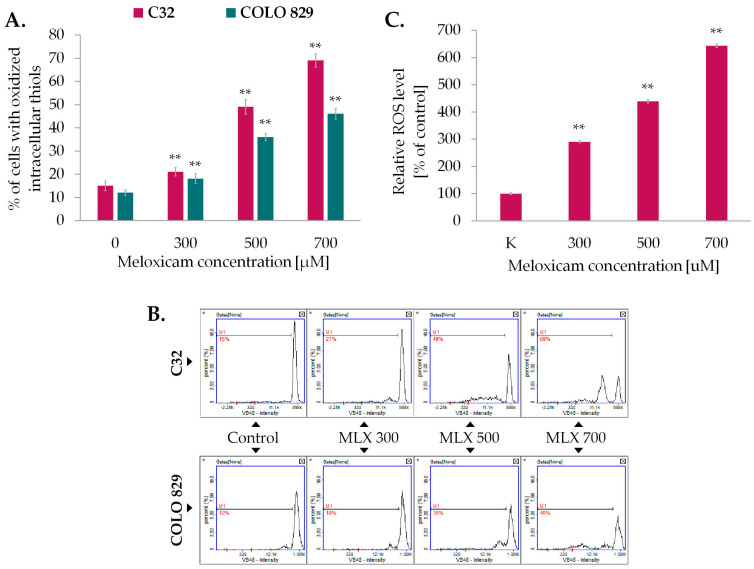
The impact of meloxicam (MLX) on oxidized thiol levels of amelanotic, C32, and mela-notic, COLO 829 melanoma (**A**), and ROS level of amelanotic melanoma (C32) (**C**). The cells were cultured for 48 h with MLX at a concentration of 300, 500, and 700 μM. Bars represent the mean ± SD (standard deviation) of 3 independent experiments, ** *p* < 0.01. Representative histograms from the analysis (**B**) depict the populations of cells with oxidized reduced thiols (M1).

**Figure 5 ijms-26-05985-f005:**
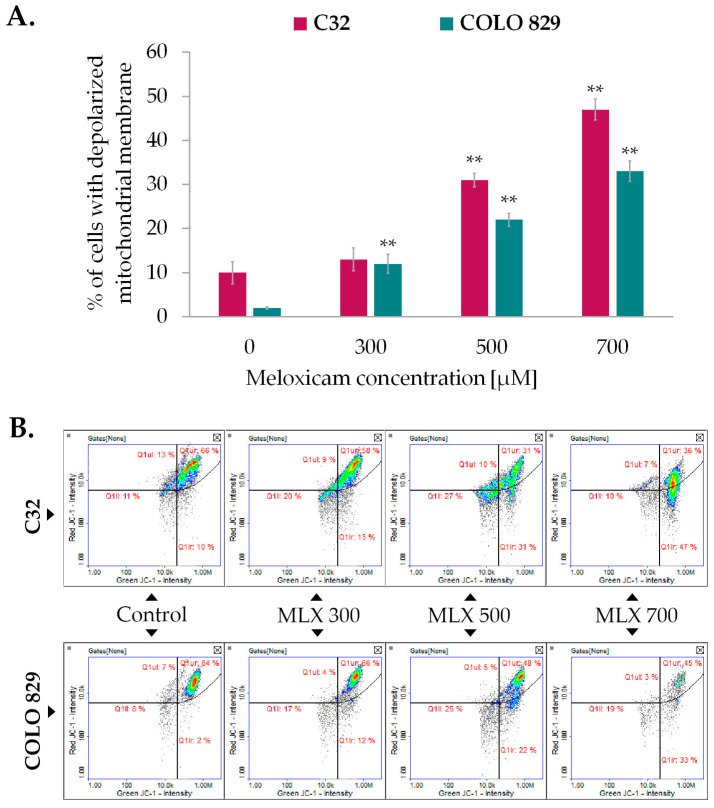
The effect of meloxicam (MLX) on mitochondrial transmembrane potential in amelanotic C32, and melanotic, COLO 829 melanoma (**A**). All analyses were performed after a 48 h incubation of the cells with MLX at a concentration of 300, 500, or 700 µM. Bar graphs show the mean value ± SD of three independent experiments; ** *p* < 0.01. The scatterplots obtained from the analysis (**B**) present the populations of cells with depolarized (Q1lr) and polarized (Q1ur) mitochondria.

**Figure 6 ijms-26-05985-f006:**
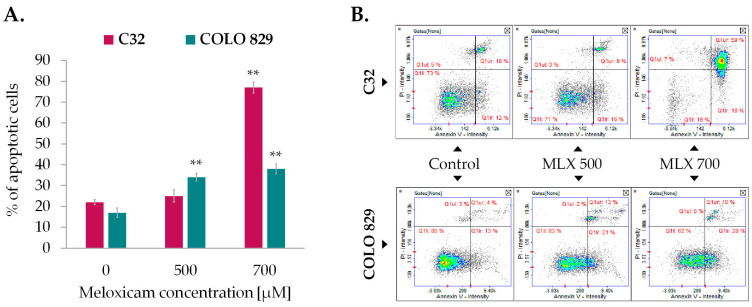
Detection of apoptosis by Annexin V assay in C32 and COLO 829 cells treated with meloxicam (MLX) in concentrations of 300 µM, 500 µM, and 700 µM (**A**). Scatter plots representing cell population are divided into quadrants: lower left—healthy cells, lower right—early apoptotic cells, upper right—late apoptotic cells, and upper left—non-apoptotic dead cells (**B**). Mean values ± SD of the percentage of apoptotic cells from three independent experiments are displayed in the bar graph; ** *p* < 0.01.

**Figure 7 ijms-26-05985-f007:**
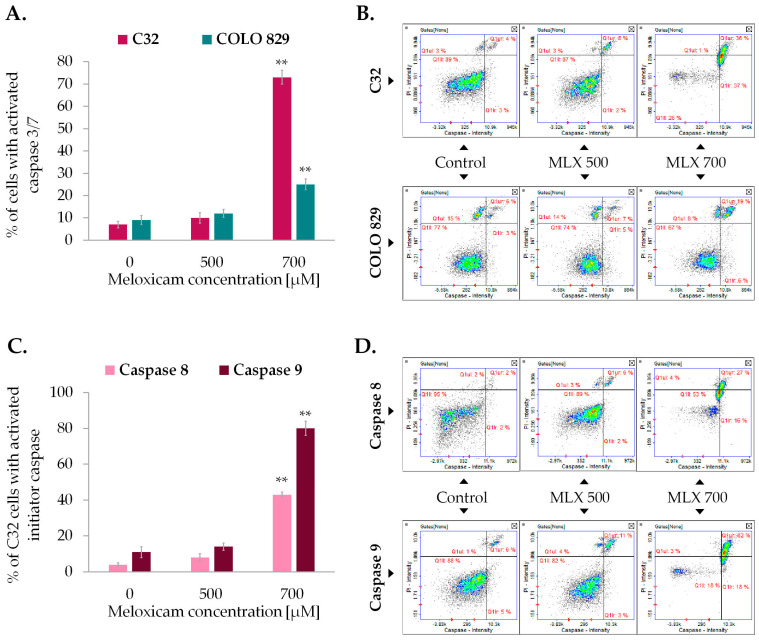
Detection of caspase 3/7 activation in amelanotic (C32) and melanotic (COLO 829) melanoma exposed to meloxicam (MLX) in concentrations of 300, 500, and 700 µM for 48 h (**A**,**C**). The assessment of the activation of initiator caspases 8 and 9 in amelanotic (C32) melanoma treated with MLX. Representative scatter plots (**B**,**D**) show cells divided into the subpopulations of cells with and without activated caspase. ** *p* < 0.01 vs. control.

**Figure 8 ijms-26-05985-f008:**
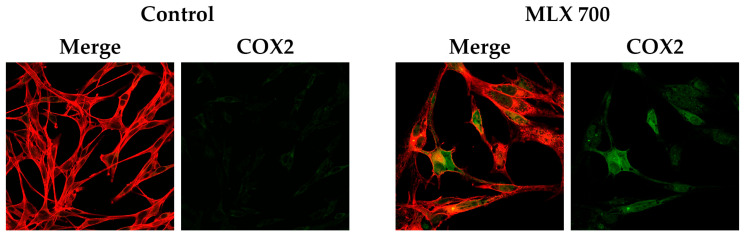
Elevated expression of COX2 in C32 cells incubated with meloxicam in a concentration of 700 µM (MLX 700). Representative confocal images presenting immunolabeled COX2 (green) and actin filaments (red).

## Data Availability

Data will be made available on request.

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
