# Peer review of "The Evaluation of Potential Anticancer Activity of Meloxicam—In Vitro Study on Amelanotic and Melanotic Melanoma"

_ijms, 2025, doi:10.3390/ijms26135985_

Round 1

Reviewer 1 Report

Comments and Suggestions for Authors

This study investigates the anticancer potential of meloxicam (MLX), a selective COX-2 inhibitor, in human melanoma cell lines—amelanotic C32 and melanotic COLO 829. MLX significantly reduced cell viability, particularly in the C32 line, and induced G2/M phase cell cycle arrest. The treatment also disrupted redox homeostasis by elevating ROS levels and oxidized thiols, resulting in mitochondrial membrane depolarization and apoptosis. Activation of apoptotic markers, including Annexin V binding and caspases 3/7, 8, and 9, confirmed involvement of both intrinsic and extrinsic apoptotic pathways. The findings suggest that MLX exerts anticancer effects through COX-2-dependent and independent mechanisms, notably redox imbalance and mitochondrial dysfunction. Its selective toxicity toward melanoma cells, while sparing normal melanocytes, underscores its potential for drug repurposing in melanoma therapy. However, further in vivo studies are essential to validate its efficacy and safety.

  1. While the authors demonstrate that MLX inhibits tumor proliferation, promotes apoptosis, and disrupts redox balance, the manuscript lacks detailed investigation into the precise molecular mechanisms underlying these effects. A mechanistic dissection—especially linking COX-2 inhibition to downstream apoptotic or redox pathways—is notably absent and should be addressed or acknowledged.
  2. Although the discussion compares MLX favorably to other agents with higher EC50 values, the EC50 for COLO 829 (270 µM) and for C32 (115 µM) remains relatively high. These values may raise concerns about potential general toxicity in in vivo models. The authors should discuss this limitation explicitly.
  3. Interestingly, the EC50 values for C32 cells do not decrease significantly across 24h, 48h, and 72h incubations (115 µM, 152 µM, and 112 µM, respectively). This plateau effect should be discussed, as it may suggest early onset saturation or limited cumulative toxicity with prolonged exposure.
  4. While the manuscript proposes COX-2-dependent activity as a mode of MLX action, there is no direct validation of COX-2 expression or functional assays (e.g., siRNA knockdown or COX-2 inhibition rescue). Incorporating such data or clarifying the absence would strengthen the mechanistic claims.
  5. Line 230: "1000 uµM" should be corrected to "1000 µM" (remove extra "u"). Additional minor grammatical and typographical errors should be reviewed throughout the manuscript for improved clarity.

Reviewer 2 Report

Comments and Suggestions for Authors

The manuscript provides a thorough in vitro investigation of meloxicam's (MLX) anticancer potential against two melanoma cell lines (amelanotic C32 and melanotic COLO 829). The study is well-designed, employs relevant methodologies, and provides mechanistic insights into the action of MLX. It explores mitochondrial dysfunction, ROS generation, apoptosis pathways, and redox homeostasis — all of which are critical in melanoma biology.

However, to enhance the scientific rigor and translational relevance of the study, improvements are recommended.

  1. The study uses MLX concentrations up to 800 µM, with EC50 values typically over 100 µM. There is no discussion about how these in vitro doses compare to achievable plasma concentrations in vivo. Recommendation includes pharmacokinetic data or literature referencing the maximal plasma levels of MLX in humans and discuss whether the tested doses are physiologically relevant.
  2. The study compares C32 (amelanotic) and COLO 829 (melanotic) melanoma lines, but it does not sufficiently discuss their backgrounds or how representative they are of clinical melanomas. Recommendation includes a brief discussion on the mutation profiles of these cell lines and how these may influence sensitivity to COX-2 inhibitors.
  3. The redox-based experiments are well-executed and show strong links between ROS levels and cell death. Improvement could be made in a discussion of how these redox alterations might interact with COX-2 signaling or whether melanin content could contribute to oxidative stress buffering.
  4. Include a brief comparison of other repurposed NSAIDs (e.g., celecoxib) and their performance in melanoma models.
  5. Referencing style and figure captions should be standardized according to the journal's formatting.
  6. Subtopics - Add purity of all reagents.
Comments on the Quality of English Language

Improve clarity in some sentences that are awkwardly structured (language editing recommended).

Correct consistent misspellings (e.g., “methastasis” → “metastasis”; “amelanoctic” → “amelanotic”).

English Suggestions:

Line 10
Original: Meloxicam (MLX), which belongs to the group of non-steroidal anti-inflammatory drugs (NSAIDs), is a preferential inhibitor of cyclooxygenase 2 (COX-2)...
Reviewed: Meloxicam (MLX), a member of the non-steroidal anti-inflammatory drugs (NSAIDs), is a preferential inhibitor of cyclooxygenase-2 (COX-2)...

Line 14
Original: The aim of the study was to investigate the potential anti-cancer activity of meloxicam against amelanotic C32 and melanotic COLO 829 melanoma cell lines.
Reviewed: This study aimed to investigate the potential anticancer activity of meloxicam against amelanotic C32 and melanotic COLO 829 melanoma cell lines.

Line 16
Original: using the WST-1 probe, analysing mitochondrial potential, reduced thiols level, annexin and caspases 3/7, 8 and 9 by imaging cytometry,
Reviewed: using the WST-1 assay and analyzing mitochondrial potential, levels of reduced thiols, annexin, and caspases 3/7, 8, and 9 by imaging cytometry,

Line 19
Original: Amelanotic melanoma, C32 was found to be more sensitive to MLX exposure by antiproliferating effect, intracellular redox homeostasis disruption, lowering the mitochondrial potential, and apoptosis induction.
Reviewed: The amelanotic melanoma C32 was more sensitive to MLX exposure, exhibiting antiproliferative effects, disruption of redox homeostasis, reduction in mitochondrial potential, and induction of apoptosis.

Line 20
Original: The results obtained offer solid molecular evidence supporting the pharmacological effects of MLX,
Reviewed: The results provide robust molecular evidence supporting the pharmacological effects of MLX,

Line 22
Original: melanoma treatment in vivo.
Reviewed: in vivo melanoma treatment.

Line 27
Original: Malignant melanoma is considered to be the most dangerous form of skin cancer due to its aggressive nature, high metastatic potential and rapid spread to other parts of the body e.g. lungs, liver, brain or bones [1,2].
Reviewed: Malignant melanoma is considered the most dangerous form of skin cancer due to its aggressive nature, high metastatic potential, and rapid spread to other organs, such as the lungs, liver, brain, or bones [1,2].

Line 31
Original: Melanocytes, as a result of environmental stimulation, e.g. intense UV exposure, the occurrence of oncogenic mutations, such as the BRAF gene mutation, or exposure to biological agents, can undergo clonal expansion and dedifferentiation, losing their features and invading deeper layers of the skin [3,4,5].
Reviewed: In response to environmental stimuli—such as intense UV exposure, oncogenic mutations (e.g., BRAF), or biological agents—melanocytes can undergo clonal expansion and dedifferentiation, losing their characteristics and invading deeper layers of the skin [3,4,5].

Line 36
Original: Accumulation of pathogenic alterations in the genome and epigenome results in disruption of intracellular homeostasis and contributes to the induction of melanoma development [7].
Reviewed: The accumulation of pathogenic alterations in the genome and epigenome disrupts intracellular homeostasis and promotes melanoma development [7].

Line 39
Original: Despite a steady global increase in knowledge of melanoma formation, classification and targeted therapy, epidemiological data remain unsatisfactory, as this cancer is the 19th most commonly reported cancer worldwide [8].
Reviewed: Despite growing global knowledge on melanoma pathogenesis, classification, and targeted therapy, epidemiological data remain unsatisfactory; melanoma ranks as the 19th most commonly reported cancer worldwide [8].

Line 41
Original: were observed in Australia, New …
Reviewed: have been reported in Australia, New …

Line 42
Original: By 2040, the global burden is expected to increase by 50% in new cases and 68% in deaths if current trends will be continued, which makes this cancer the leading cause of death for patients
Reviewed: If current trends continue, the global burden of melanoma is expected to rise by 50% in new cases and 68% in deaths by 2040, potentially making it a leading cause of cancer-related mortality.

Line 45
Original: The treatment of melanoma has evolved significantly, with immunotherapy and targeted therapy leading to improved survival rates.
Reviewed: Melanoma treatment has advanced significantly, with immunotherapies and targeted therapies contributing to improved survival rates.

Line 46
Original: Surgery remains essential for early-stage melanoma, but advanced cases require combination of therapies.
Reviewed: Surgery remains critical for early-stage melanoma, while advanced cases often require combination therapies.

Line 47
Original: Unfortunatelly, in spite of advancements, challenges that include resistance and toxicity occurrence, lead to conducting research into novel treatment strategies [11].
Reviewed: Unfortunately, despite these advancements, challenges such as drug resistance and toxicity persist, driving the search for novel treatment strategies [11].

Line 50
Original: It is believed that one of the pathways of melanoma development is related to the COX-2 enzyme (cyclooxygenase 2),
Reviewed: One proposed pathway in melanoma development involves the COX-2 enzyme (cyclooxygenase-2),

Line 51
Original: which is induced by external factors such as UV radiation and TNF-α (tumor necrosis factor α) [12].
Reviewed: which can be induced by external factors such as UV radiation and tumor necrosis factor-alpha (TNF-α) [12].

Line 52
Original: Additionally It enhances the depth of invasion, increases the risk of methastasis leading to poorer survival rates [13].
Reviewed: Additionally, it increases invasion depth and the risk of metastasis, resulting in poorer survival rates [13].

Line 54
Original: Therefore, the inhibition of COX-2 enzyme, could represent a promising strategy to inhibit the growth of melanoma cells.
Reviewed: Therefore, inhibition of the COX-2 enzyme may represent a promising strategy to suppress melanoma cell growth.

Line 56
Original: Meloxicam (MLX), belonging to the group of non-steroidal anti-inflammatory drugs (NSAIDs), is a selective inhibitor of COX-2 enzyme and is widely used for its anti-inflammatory and analgesic effects.
Reviewed: Meloxicam (MLX), a non-steroidal anti-inflammatory drug (NSAID), is a selective COX-2 inhibitor widely used for its anti-inflammatory and analgesic effects.

Line 58
Original: Previous studies have confirmed that MLX also possesses anti-cancer properties and reduces cancer cell proliferation, inhibits angiogenesis and promotes programmed cell death [14].
Reviewed: Previous studies have shown that MLX also exhibits anticancer properties by reducing cancer cell proliferation, inhibiting angiogenesis, and promoting programmed cell death [14].

Line 60
Original: These effects have been demonstrated in lung cancer, breast cancer, colorectal cancer and glioma [15–18].
Reviewed: These effects have been demonstrated in lung, breast, and colorectal cancers, as well as glioma [15–18].

Line 62
Original: However, the molecular mechanisms of MLX action on melanoma cells remain poorly understood and require more detailed research.
Reviewed: However, the molecular mechanisms underlying MLX action on melanoma cells remain poorly understood and warrant further investigation.

Line 69
Original: the study was
Reviewed: this study was

Line 70
Original: on the possible therapeutic applications of this compound in inhibiting melanoma development.
Reviewed: on its potential therapeutic application in inhibiting melanoma progression.

Line 74
Original: The results of the WST-1 assay showed that MLX had an inhibitory effect on the proliferation of both melanoma cell lines,
Reviewed: The WST-1 assay results showed that MLX inhibited the proliferation of both melanoma cell lines,

Line 75
Original: more sensitive to its action than the melanotic
Reviewed: more sensitive than the melanotic

Line 77
Original: The viability of C32 cells treated with MLX at concentrations of 2.5 mM and 5 mM decreased to 79.6% and 42.6% respectively,
Reviewed: The viability of C32 cells treated with 2.5 mM and 5 mM MLX decreased to 79.6% and 42.6%, respectively,

Line 85
Original: In the case of C32 cells, a statistically significant decrease in thiol groups was observed for both 2.5 mM and 5 mM MLX, with the highest decrease seen at the higher concentration (Figure 2A).
Reviewed: In C32 cells, a statistically significant reduction in thiol groups was observed at both 2.5 mM and 5 mM MLX, with the greatest decrease at the higher concentration (Figure 2A).

Line 87
Original: For COLO 829 cells, only treatment with 5 mM MLX caused a statistically significant decrease in the level of thiol groups (Figure 2B).
Reviewed: In COLO 829 cells, only 5 mM MLX caused a statistically significant decrease in thiol group levels (Figure 2B).

Line 91
Original: The results of the assessment of ROS levels in MLX-treated cells showed a statistically significant increase in ROS levels only in the amelanotic cell line C32.
Reviewed: Assessment of ROS levels in MLX-treated cells revealed a statistically significant increase only in the amelanotic C32 cell line.

Line 93
Original: MLX significantly increased the fluorescence intensity of the H2DCFDA probe by 5.2% and 10.2%, respectively (Figure 3A).
Reviewed: MLX increased H2DCFDA fluorescence intensity by 5.2% and 10.2%, respectively (Figure 3A).

Line 113
Original: For COLO 829 cells
Reviewed: In COLO 829 cells

Line 115
Original: The increase in apoptosis was dose-dependent in both cell lines.
Reviewed: Apoptosis increased in a dose-dependent manner in both cell lines.

Line 121
Original: The study assessed the effect of MLX on the activity of caspases 3/7, 8 and 9 in both melanoma cell lines.
Reviewed: This study assessed MLX's effect on caspase 3/7, 8, and 9 activity in both melanoma cell lines.

Line 123
Original: The results showed that MLX significantly increased the activity of all three caspases in C32 cells in a dose-dependent manner (Figure 6A).
Reviewed: MLX significantly increased the activity of all three caspases in C32 cells in a dose-dependent manner (Figure 6A).

Line 136
Original: The presence of melanin in COLO 829 cells may reduce the effectiveness of MLX by acting as a free radical scavenger.
Reviewed: The melanin present in COLO 829 cells may reduce MLX effectiveness by scavenging free radicals.

Line 143
Original: MLX has been shown to exert antiproliferative effects in various cancers, including lung, breast, and colon cancer [15–18].
Reviewed: MLX has demonstrated antiproliferative effects in lung, breast, and colorectal cancers [15–18].

Line 147
Original: The greater sensitivity
Reviewed: The higher sensitivity

Line 162
Original: Our findings of decreased ΔΨm in C32 but not in COLO 829 cells further support the selective action of MLX.
Reviewed: The observed reduction in ΔΨm in C32 cells, but not in COLO 829 cells, further supports the selective activity of MLX.

Line 187
Original: Cells were treated for 24 h with meloxicam at indicated concentrations.
Reviewed: Cells were treated with meloxicam for 24 h at the indicated concentrations..

Line 195
Original: The decrease in ΔΨm indicates mitochondrial dysfunction.
Reviewed: A decrease in ΔΨm indicates mitochondrial dysfunction.

Line 206
Original: considered to be the third most common form of skin cancer.
Reviewed: considered the third most common form of skin cancer.

Line 208
Original: continue to face a poor prognosis, and high risk of mortality – approx. 75%.
Reviewed: still face a poor prognosis and a high mortality rate—approximately 75%.

Line 218
Original: highly expressed in malignant melanoma and may be associated with disease progression.
Reviewed: overexpressed in malignant melanoma and may contribute to disease progression.

Line 222
Original: In the first step of the study, the cytotoxic potential of MLX against melanoma cells was assessed.
Reviewed: In the initial phase of the study, the cytotoxic potential of MLX against melanoma cells was assessed.

Line 223
Original: suppressed cell proliferation in a concentration-dependent manner.
Reviewed: inhibited cell proliferation in a concentration-dependent manner.

Line 224
Original: 72h incubation, respectively.
Reviewed: 72 h of incubation, respectively.

Line 226
Original: 48h and 72h.
Reviewed: 48 h, and 72 h, respectively.

Line 230
Original: This indicates that the drug is selective towards tumour cells.
Reviewed: These findings suggest that MLX exhibits selective cytotoxicity toward tumor cells.

Line 231
Original: It is noteworthy that the EC50 values obtained are similar to, or even lower than, other drugs tested in cellular models of melanoma treatment...
Reviewed: Notably, the EC50 values obtained are comparable to, or even lower than, those of other drugs tested in melanoma cell models...

Line 247
Original: High levels of ROS can cause cell cycle arrest, consequently leading to apoptosis [33].
Reviewed: Elevated ROS levels can induce cell cycle arrest, thereby triggering apoptosis [33].

Line 248
Original: To counteract elevated intracellular ROS levels, cancer cells have both low-molecular scavengers, which include glutathione (GSH), and specific antioxidant enzymes - superoxide dismutase, glutathione peroxidase and catalase - whose transcription is activated as a result of high levels of ROS in cells [34].
Reviewed: To counteract elevated intracellular ROS, cancer cells utilize both low-molecular-weight scavengers such as glutathione (GSH), and specific antioxidant enzymes—superoxide dismutase, glutathione peroxidase, and catalase—whose transcription is upregulated in response to high ROS levels [34].

Line 260
Original: The percentage of cells with high level of oxidized thiols was about 70% for C32 and 50% for COLO 829 melanoma cells treated with 700 µM of MLX.
Reviewed: Approximately 70% of C32 cells and 50% of COLO 829 cells treated with 700 µM MLX showed high levels of oxidized thiols.

Line 270
Original: For ROS content, MLX at 1000 µM was found to cause an approx. 50% increase in free radical density in fibroblasts, while in melanocytes the increase was about 80% [37].
Reviewed: Regarding ROS content, MLX at 1000 µM caused an approximately 50% increase in free radical density in fibroblasts and about 80% in melanocytes [37].

Line 273
Original: The data obtained indicate that MLX has a good safety profile with respect to normal skin cells, causing greater changes in melanoma cells at the lower dose used.
Reviewed: These findings indicate that MLX has a favorable safety profile for normal skin cells, inducing more pronounced effects in melanoma cells even at lower doses.

Line 275
Original: Similar results were obtained in studies using other NSAIDs drugs, diclofenac and piroxicam on melanoma line SK-MEL-5R has been shown to increase ROS
Reviewed: Similar results have been reported with other NSAIDs, such as diclofenac and piroxicam, which increased ROS levels in the SK-MEL-5R melanoma cell line

Line 282
Original: The intrinsic pathway, dependent on mitochondria, involves the release of cytochrome c, which forms the apoptosome with Apaf-1 and procaspase-9, leading to the activation of caspase-9 and effector caspases - 3 and 7.
Reviewed: The intrinsic, mitochondria-dependent pathway involves cytochrome c release, which forms the apoptosome with Apaf-1 and procaspase-9, leading to activation of caspase-9 and effector caspases 3 and 7.

Line 291
Original: MLX was shown to significantly increase annexin V-positive cells, although a higher proportion of apoptotic cells was found in amelanotic melanoma in the concentration of 700 µM.
Reviewed: MLX significantly increased the proportion of annexin V-positive cells, with a higher percentage of apoptotic cells observed in amelanotic melanoma at 700 µM.

Line 295
Original: The drug in concentration of 700 µM increased the percentage of cells with depolarized mitochondrial membrane up to approx. 47% in amelanotic C32 melanoma and ca. 33% in the case of COLO 829 cell line.
Reviewed: At a concentration of 700 µM, the drug increased the percentage of cells with depolarized mitochondrial membranes to approximately 47% in amelanotic C32 melanoma and about 33% in COLO 829 cells.

Author Response

We would like to thank the Reviever for the good reception of the presented manuscript, all insightful suggestions and comments. We have revised the manuscript following the Reviewer’s remarks.

COMMENT: The study uses MLX concentrations up to 800 µM, with EC50 values typically over 100 µM. There is no discussion about how these in vitro doses compare to achievable plasma concentrations in vivo. Recommendation includes pharmacokinetic data or literature referencing the maximal plasma levels of MLX in humans and discuss whether the tested doses are physiologically relevant.

RESPONSE: The concentrations of meloxicam applied in this in vitro study exceed those typically detected in plasma or used in in vivo settings. This discrepancy reflects the specific requirements of in vitro experimentation, where higher drug levels are often essential to produce measurable cellular responses. Consequently, translating these findings into in vivo models—whether in animals or clinical patients—necessitates appropriate adjustments to the effective dose used in vitro.

In the manuscript Authors added information:

In in vitro studies, meloxicam is often used at higher concentrations than those typically administered in vivo. This adjustment is necessary due to the specific nature of in vitro conditions, where higher drug doses are required to produce observable and measurable effects on isolated cells or tissues. Unlike in vivo systems, where pharmacokinetics and systemic metabolism influence drug availability, in vitro environments lack these dynamic processes, potentially reducing the apparent potency of the compound. Therefore, to achieve relevant biological responses and investigate the mechanisms of action effectively, elevated doses of meloxicam are applied. However, findings from such studies must be carefully interpreted, as translating these results to in vivo settings in animals or humans will require dose modifications to account for physiological and safety considerations.

COMMENT: The study compares C32 (amelanotic) and COLO 829 (melanotic) melanoma lines, but it does not sufficiently discuss their backgrounds or how representative they are of clinical melanomas. Recommendation includes a brief discussion on the mutation profiles of these cell lines and how these may influence sensitivity to COX-2 inhibitors.

RESPONSE: Following the Reviewer's recommendation, an excerpt from the discussion regarding the impact of differences in melanotic and amelanotic melanoma mutations and their potential impact on COX-2 sensitivity has been added.

Noteworthy, melanotic and amelanotic melanoma cell lines differ significantly in their pigmentation levels, reflecting cellular differentiation and different molecular signatures. These differences may have a critical impact on the cellular response to both targeted therapies and COX-2 inhibitors. Melanotic melanoma cells often retain functional melanogenesis pathways and carry mutations commonly found in cutaneous melanoma, such as BRAF V600E, NRAS or NF1 [49,50]. These mutations promote sustained activation of the MAPK signalling cascade, often resulting in elevated COX-2 expression [61]. Consequently, melanotic lines tend to exhibit higher baseline levels of COX-2 and are generally more susceptible to COX-2 inhibition, which may inhibit proliferation and inflammation-related survival pathways [62]. In contrast, amelanotic melanoma cells tend to be less differentiated and are often characterised by reduced expression of pigmentation-related genes (e.g. MITF, TYR) and alternative oncogenic alterations. These may include abnormal activation of the PI3K/AKT pathway or loss-of-function mutations in tumour suppressors such as TP53, potentially leading to reduced COX-2 expression and reduced sensitivity to its inhibition [49]. Nevertheless, some amelanotic cell lines may exhibit compensatory up-regulation of inflammatory mediators, making them partially sensitive to COX-2 blockade, especially when combined with other targeted or immunomodulatory agents [62]. In summary, mutation diversity and differentiation status of melanoma cells play a key role in modulating COX-2 expression and activity, which in turn influences the therapeutic efficacy of COX-2 inhibitors. These insights support a precision oncology approach to selectively use COX-2-targeting strategies based on tumour subtype and molecular profile.

  1. Soengas, M.S.; Lowe, S.W. Apoptosis and melanoma chemoresistance. Oncogene 2003, 22(20), 3138-51. doi: 10.1038/sj.onc.1206454.
  2. Piskounova, E.; Agathocleous, M.; Murphy, M.M.; Hu, Z.; Huddlestun, S.E.; Zhao, Z.; Leitch, A.M.; Johnson, T.M.; DeBerardinis, R.J.; Morrison, S.J. Oxidative stress inhibits distant metastasis by human melanoma cells. Nature 2015, 527(7577), 186-91. doi: 10.1038/nature15726.
  3. Denkert, C.; Köbel, M.; Pest, S.; Koch, I.; Berger, S.; Schwabe, M.; Siegert, A.; Reles, A.; Kloster-halfen, B.; Hauptmann, S. Expression of cyclooxygenase 2 is an independent prognostic factor in human ovarian carcinoma. Am J Pathol 2002, 160(3), 893-903. doi: 10.1016/S0002-9440(10)64912-7.
  4. Dhawan, P.; Richmond, A. Role of CXCL1 in tumorigenesis of melanoma. J of Leuk Biol 2002, 72(1), 9–18. https://doi.org/10.1189/jlb.72.1.9

COMMENT: The redox-based experiments are well-executed and show strong links between ROS levels and cell death. Improvement could be made in a discussion of how these redox alterations might interact with COX-2 signaling or whether melanin content could contribute to oxidative stress buffering.

RESPONSE: The Authors added a section in the Discussion section on the interaction between COX-2 and redox homeostasis.

Inhibition of COX-2 is molecularly associated with the induction of apoptosis and the disruption of redox homeostasis through several interconnected pathways. COX-2 catalyzes the conversion of arachidonic acid to prostaglandins, particularly prostaglandin E2 (PGE2), which promotes cell survival by activating pro-proliferative and anti-apoptotic signaling cascades, including the PI3K/Akt and ERK1/2 MAPK pathways [54]. Suppression of COX-2 leads to decreased PGE2 synthesis, resulting in downregulation of Bcl-2 family anti-apoptotic proteins (e.g., Bcl-2, Bcl-xL) and upregulation of pro-apoptotic factors such as Bax and caspase-3 activation, thereby initiating the intrinsic (mitochondrial) apoptotic pathway [55]. Moreover, COX-2 inhibition significantly impacts the redox state of the cell. Under physiological conditions, COX-2 modulates ROS levels and affects the expression of antioxidant enzymes such as superoxide dismutase (SOD) and glutathione peroxidase (GPx) [56]. Inhibition of COX-2 disrupts this balance, often resulting in excessive accumulation of reactive oxygen species (ROS), which damages lipids, proteins, and DNA, and promotes mitochondrial membrane depolarization, cytochrome c release, and subsequent activation of caspase-dependent apoptosis [57]. Additionally, elevated ROS levels can modulate redox-sensitive transcription factors, including NF-κB and AP-1, which control the expression of genes related to inflammation, survival, and apoptosis. COX-2 inhibition may attenuate NF-κB activation, leading to reduced transcription of survival genes and a shift toward apoptotic cell fate [58]. Thus, COX-2 inhibitors such as meloxicam not only suppress inflammatory signaling but also promote apoptosis and oxidative stress, contributing to altered redox homeostasis and cell death in both pathological and therapeutic contexts.

  1. Greenhough, A.; Smartt, H.J.; Moore, A.E.; Roberts, H.R.; Williams, A.C.; Paraskeva, C.; Kaidi, A. The COX-2/PGE2 pathway: key roles in the hallmarks of cancer and adaptation to the tumour microenvironment. Carcinogenesis 2009, 30(3), 377-86. doi: 10.1093/carcin/bgp014.
  2. Yamamoto, Y.; Yin, M.J.; Lin, K.M.; Gaynor, R.B. Sulindac inhibits activation of the NF-kappaB pathway. J Biol Chem 1999, 274(38), 27307-14. doi: 10.1074/jbc.274.38.27307.
  3. Zhou, Y.; Yao, Y.; Chen, J.; Chen, J.; Chen, Z. Inhibition of cyclooxygenase-2 induces apoptosis and inhibits proliferation in human bladder cancer cells. Oncology Reports 2005, 14(4), 879–884.
  4. Huang, H.; Zhang, S.; Li, Y.; Liu, Z.; Mi, L.; Cai, Y.; Wang, X.; Chen, L.; Ran, H.; Xiao, D.; Li, F.; Wu, J.; Li, T.; Han, Q.; Chen, L.; Pan, X.; Li, H.; Li, T.; He, K.; Li, A.; Zhang, X.; Zhou, T.; Xia, Q.; Man, J. Suppression of mitochondrial ROS by prohibitin drives glioblastoma progression and therapeutic resistance. Nat Commun 2021, 12(1), 3720. doi: 10.1038/s41467-021-24108-6.

58.Tsujii, M.; DuBois, R.N. Alterations in cellular adhesion and apoptosis in epithelial cells overe-xpressing prostaglandin endoperoxide synthase 2. Cell 1995, 83(3), 493-501. doi: 10.1016/0092-8674(95)90127-2.

COMMENT: Include a brief comparison of other repurposed NSAIDs (e.g., celecoxib) and their performance in melanoma models.

RESPONSE: The Authors supplemented the Discussion section with a Reviewer recommended discussion of the effects of other NSAIDs on melanoma cells.

Other COX-2 inhibitors, such as celecoxib, can reduce melanoma cell via-bility, proliferation, and metastatic potential in preclinical models [48]. The anti-melanoma effect of NSAIDs is thought to be mediated not only through COX-2 inhibition but also via COX-independent pathways, in-cluding modulation of mitochondrial apoptosis and suppression of Akt signaling [49]. Additionally, NSAIDs may enhance the effectiveness of ex-isting therapies, including immune checkpoint inhibitors, by reducing prostaglandin-mediated immunosuppression in the tumor microenviron-ment [50].

  1. Xin, B.; Yokoyama, Y.; Shigeto, T.; Mizunuma, H. Anti-tumor effect of non-steroidal an-ti-inflammatory drugs on human ovarian cancers. Pathol Oncol Res 2007, 13(4), 365-9. doi: 10.1007/BF02940318.
  2. Soengas, M.S.; Lowe, S.W. Apoptosis and melanoma chemoresistance. Oncogene 2003, 22(20), 3138-51. doi: 10.1038/sj.onc.1206454.
  3. Piskounova, E.; Agathocleous, M.; Murphy, M.M.; Hu, Z.; Huddlestun, S.E.; Zhao, Z.; Leitch, A.M.; Johnson, T.M.; DeBerardinis, R.J.; Morrison, S.J. Oxidative stress inhibits distant metastasis by human melanoma cells. Nature 2015, 527(7577), 186-91. doi: 10.1038/nature15726.

COMMENT: Referencing style and figure captions should be standardized according to the journal's formatting.

RESPONSE: Referencing style and figure captions have been corrected to comply with the journal’s requirements.

COMMENT: Subtopics - Add purity of all reagents.

RESPONSE: The Authors added in matherials section available informations about agents purity. Regarding the purity of materials purchased from Roche, ChemoMetec, Biotum, ImmunoChemistry Technologies do not publicly disclose exact purity percentages for this reagents. However, the reagents have been purchased from suppliers guaranteeing that they can be used in biochemical analyses and are of laboratory standard.

Materials and methods section:

Meloxicam was obtained from Boehringer Ingelheim (purity ≥98 % (HPLC)) (Budapest, Hungary). Dulbecco’s Modified EagleMedium (DMEM), Roswell Park Memorial Institute (RPMI) 1640 medium, Trypsin/EDTA (0.25%/0.02%), and Fetal Bovine Serum (FBS) were obtained from PAN-Biotech GmbH (Aidenbach, Germany). H2DCFDA reagent, Dulbecco’s phosphate-buffered saline (DPBS), Penicillin-Streptomycin solution (10,000 U/mL) (purity: ≥97 – ≥99 %) HPLC)), anti-rabbit secondary antibody Alexa Fluor 488 conjugate were purchased from Thermo Fisher Scientific Inc. (Waltham, MA, USA). WST-1 cell proliferation reagent was obtained from Roche GmbH (Mannheim, Germany). Solution 3 (DAPI 1 µg/mL, triton X-100 0.1%), Solution 5 (VitaBright-48 400 µg/mL, propidium iodide 500 µg/mL, acridine orange 1.2 µg/mL), Solution 7 (JC-1 dye 200 µg/mL), Solution 8 (DAPI 1 µg/mL), Solution 15 (Hoechst 33342 500 µg/mL), Solution 16 (propidium iodide 500 µg/mL), Via1-Cassettes, and NC-Slides A2 and A8 were purchased from ChemoMetec (Lillerød, Denmark). Annexin V-CF488A was obtained from Biotium (Fremont, CA USA). 10x Annexin V binding buffer was obtained from BioVision Inc. (Milpitas, CA, USA). Green Fluorescent FAM-FLICA Caspase-3/7 Assay Kit, Green Fluorescent FAM-FLICA Caspase 8 Assay Kit, and Green Fluorescent FAM-FLICA Caspase 9 Assay Kit were obtained from ImmunoChemistry Technologies (Bloomington, MN, USA). COX2 (D5H5) XP Rabbit mAb was obtained from Cell Signaling Technology (Danvers, MA, USA). Bovine Serum Albumin (BSA) and Phalloidin-Atto565 were purchased from Sigma Aldrich(St. Louis, MO, USA). Fluorescence mounting medium was Dako.

The Authors ensure that the Reviewer's suggested changes in English have been carried out.

Reviewer 3 Report

Comments and Suggestions for Authors

In this interesting article, the authors investigated the antitumor potential of meloxicam
against amelanotic and melanotic melanoma cell lines. The present research has the necessary
depth of analysis and significant scientific merit, leading to a better understanding of the anticancer potential of meloxicam in the treatment of malignant melanoma. The Manuscript is well written, with a lot of interesting results, and with detailed discussion. However, some revisions should be addressed to improve the quality of the manuscript:

1. In the Introduction section, describe the current scientific knowledge regarding specific
anticancer agents used in the clinical treatment of malignant melanoma, along with the
corresponding references.
2. In the Introduction section, please add the paragraph regarding the physico-chemical
properties of meloxicam, as well as the Figure displaying the chemical structure of
meloxicam.
3. Are there any published studies that have examined the antitumor potential of meloxicam
or its derivatives in treating malignant melanoma? If so, please cite the relevant studies. If
not, emphasize that fact as an important aspect of the novelty of the current research.
4. Was the WST-1 probe conducted in triplicate? This is not emphasized in the Materials and
Methods section. Please specify.
5. Page 2, Line 86: Correct the misspelled word “synthesise” to "synthesize."
6. On what basis was the meloxicam dosage concentration interval chosen in cell viability
tests? Was it based on the design of previously published studies or on laboratory
experience? Please indicate clearly in the text.
7. In the title of Figure 2, the explanation regarding Figure 2C is completely missing. Please
add it to the Figure caption. In addition, Figure 2C is quite blurred. Please increase the
figure resolution if possible.
8. Page 4, Line 134 and Page 5, Line 143: Is it Figures 3A and 3C rather than Figures 2A and
2C?
9. The histograms within Figures 3B and 4B are very difficult to read. Please increase the
resolution of these Figures.
10. Please indicate clearly what is presented in Figure 4A. The caption of Figure 4 is a little bit
confusing because the letter “a” and its explanation are missing.
11. Please change the title of Subsection 2.5 to sound more general. For example: “Assessment
of meloxicam-induced apoptosis in melanoma cells”.
12. Page 6, Line 178: Is it Figure 5A rather than Figure 4A?
13. The Scatter plots within Figure 5B are very difficult to read and understand. Please increase
the resolution significantly to enable their interpretation.
14. Page 7, Line 193: Is it Figure 6 rather than Figure 5?
15. At the end of the section Conclusions, please clearly indicate the limitations of the present
research and future research directions regarding the antitumor potential of meloxicam in
the treatment of melanoma and other malignancies.

Author Response

We would like to thank the Reviewer for the assessment of our manuscript. We are grateful for the comments and advices. According to the suggestions, we introduced following changes and corrections to the paper. All edited fragments of the manuscript are to be founds in bold.

COMMENT: In the Introduction section, describe the current scientific knowledge regarding specific anticancer agents used in the clinical treatment of malignant melanoma, along with the corresponding references.

RESPONSE: Following the Reviewer's suggestion, the Authors have added information in the Introduction section on current drugs for the treatment of malignant melanoma. The added section has been highlighted in bold.

For patients with BRAF V600 mutations, targeted therapies such as BRAF inhibitors (e.g., vemurafenib, dabrafenib) in combination with MEK inhibitors (e.g., trametinib, cobimetinib) have demonstrated improved progression-free and overall survival rates [12]. Immunotherapy, particularly immune checkpoint inhibitors like anti-PD-1 (nivolumab, pembrolizumab) and anti-CTLA-4 (ipilimumab), has revolutionized the treatment landscape by enhancing the host immune response against tumor cells [13]. Combination immunotherapy (e.g., nivolumab plus ipilimumab) has shown increased efficacy, albeit with higher toxicity profiles [14]. Additionally, adjuvant therapy with immune checkpoint inhibitors has been established as a standard for patients with resected stage III and IV melanoma [15]. Unfortunately, despite these advancements, challenges such as drug resistance and toxicity persist, driving the search for novel treatment strategies [11].

  1. Natarelli, N.; Aleman, S.J.; Mark, I.M.; Tran, J.T.; Kwak, S.; Botto, E.; Aflatooni, S.; Diaz, M.J.; Lipner, S.R. A Review of Current and Pipeline Drugs for Treatment of Melanoma. Pharmaceuticals 2024, 17, 214. https://doi.org/10.3390/ph17020214
  2. Long ,G.V.; Stroyakovskiy, D.; Gogas, H.; Levchenko, E.; de Braud, F.; Larkin, J.; Garbe, C.; Jouary, T.; Hauschild, A.; Grob, J.J.; Chiarion-Sileni, V.; Lebbe, C.; Mandal, M.; Millward, M.; Arance, A.; Bondarenko, I.; Haanen, J.B.; Hansson, J.; Utikal, J.; Ferraresi, V.; Kovalenko, N.; Mohr, P.; Probachai, V.; Schadendorf, D.; Nathan, P.; Robert, C.; Ribas, A.; DeMarini, D.J.; Irani, J.G.; Swann, S.; Legos, J.J.; Jin, F.; Mookerjee, B.; Flaherty, K. Dabrafenib and trametinib versus dabrafenib and placebo for Val600 BRAF-mutant melanoma: a multicentre, double-blind, phase 3 randomised controlled trial. Lancet 2015, 386(9992), 444-51. doi: 10.1016/S0140-6736(15)60898-4.
  3. Robert, C.; Long, G.V.; Brady, B.; Dutriaux, C.; Maio, M.; Mortier, L.; Hassel, J.C.; Rutkowski, P.; McNeil, C.; Kalinka-Warzocha, E.; Savage, K.J.; Hernberg, M.M.; Lebb, C.; Charles, J.; Mihalcioiu, C.; Chiarion-Sileni, V.; Mauch, C.; Cognetti, F.; Arance, A.; Schmidt, H.; Schadendorf, D.; Gogas, H.; Lundgren-Eriksson, L.; Horak, C.; Sharkey, B.; Waxman, I.M.; Atkinson, V.; Ascierto, P.A. Nivolumab in previously untreated melanoma without BRAF mutation. N Engl J Med 2015, 372(4), 320-30. doi: 10.1056/NEJMoa1412082.
  4. Larkin, J.; Chiarion-Sileni, V.; Gonzalez, R.; Grob, J.J.; Cowey, C.L.; Lao, C.D.; Schadendorf, D.; Dummer, R.; Smylie, M.; Rutkowski, P.; Ferrucci, P.F.; Hill, A.; Wagstaff, J.; Carlino, M.S.; Haanen, J.B.; Maio, M.; Marquez-Rodas, I.; McArthur, G.A.; Ascierto, P.A.; Long, G.V.; Callahan, M.K.; Postow, M.A.; Grossmann, K.; Sznol, M.; Dreno, B.; Bastholt, L.; Yang, A.; Rollin, L.M.; Horak, C.; Hodi, F.S.; Wolchok, J.D. Combined Nivolumab and Ipilimumab or Monotherapy in Untreated Melanoma. N Engl J Med 2015, 373(1), 23-34. doi: 10.1056/NEJMoa1504030.
  5. Eggermont, A.M.M.; Blank, C.U.; Mandala, M.; Long, G.V.; Atkinson, V.; Dalle, S.; Haydon, A.; Lichinitser, M.; Khattak, A.; Carlino, M.S.; Sandhu, S.; Larkin, J.; Puig, S.; Ascierto, P.A.; Rutkowski, P.; Schadendorf, D.; Koornstra, R.; Hernandez-Aya, L.; Maio, M.; van den Eertwegh, A.J.M.; Grob, J.J.; Gutzmer, R.; Jamal, R.; Lorigan, P.; Ibrahim, N.; Marreaud, S.; van Akkooi, A.C.J.; Suciu, S.; Robert, C. Adjuvant Pembrolizumab versus Placebo in Resected Stage III Melanoma. N Engl J Med 2018, 378(19), 1789-1801. doi: 10.1056/NEJMoa1802357.

COMMENT: In the Introduction section, please add the paragraph regarding the physico-chemical properties of meloxicam, as well as the Figure displaying the chemical structure of meloxicam.

RESPONSE:  The Authors added suggested informations about physico-chemical properties of meloxicam and its chemical structure in the introduction section.

Meloxicam (MLX) is a selective COX-2 inhibitor, which belongs to the group of non-steroidal anti-inflammatory drug (NSAIDs). Chemical structure of MLX is presented on Figure 1. The drug exhibits poor aqueous solubility, with a solubility of approximately 7.15 μg/mL in water at 25°C, which presents challenges in formulation and bioavailability [18]. MLX is a weak acid with a pKa of approximately 1.1 (enolic group) and 4.2 (carboxamide group), contributing to its pH-dependent solubility profile [19]. The drug is lipophilic, exhibiting a logP value ca. 3.4, facilitating its membrane permeability and oral absorption. Meloxicam is highly bound to plasma proteins (~99%) and demonstrates a relatively long half-life of 15–20 hours, allowing for once-daily dosing [20]. These physicochemical properties influence its pharmacokinetics, efficacy, and formulation strategies, including the development of solubility-enhancing delivery systems such as solid dispersions and nanocrystals.

Figure 1. Chemical structure of meloxicam.

References:

  1. Shende, P.K.; Gaud, R.S.; Bakal, R.; Patil, D. Effect of inclusion complexation of meloxicam with β-cyclodextrin- and β-cyclodextrin-based nanosponges on solubility, in vitro release and stability studies. Colloids Surf B Biointerfaces 2015, 136, 105-10. doi: 10.1016/j.colsurfb.2015.09.002.
  2. Loftsson, T.; Brewster, M.E. Pharmaceutical applications of cyclodextrins: basic science and product development. J Pharm Pharmacol 2010, 62(11), 1607-21. doi: 10.1111/j.2042-7158.2010.01030.x.
  3. Davies, N.M.; Skjodt, N.M. Clinical pharmacokinetics of meloxicam. A cyclo-oxygenase-2 preferential nonsteroidal anti-inflammatory drug. Clin Pharmacokinet 1999, 36(2), 115-26. doi: 10.2165/00003088-199936020-00003.

COMMENT: Are there any published studies that have examined the antitumor potential of meloxicam or its derivatives in treating malignant melanoma? If so, please cite the relevant studies. If not, emphasize that fact as an important aspect of the novelty of the current research.

RESPONSE: As there are currently no reports in the literature to assess the effect of meloxicam on malignant melanoma cells, the Authors described this fact in the introduction section.

To date, there are no published studies that have specifically examined the antitumor potential of meloxicam or its derivatives in the treatment of malignant melanoma. While meloxicam has demonstrated anticancer activity in other tumor types—such as non-small cell lung cancer and colorectal cancer—its effects on melanoma remain entirely unexplored in the scientific literature. This notable absence highlights a clear gap in current oncological research and underscores the novelty of the present study. By investigating meloxicam in the context of malignant melanoma, this research addresses an unmet need and explores a previously uncharted therapeutic avenue, potentially contributing valuable insights into melanoma treatment strategies.

COMMENT: Was the WST-1 probe conducted in triplicate? This is not emphasized in the Materials and Methods section. Please specify.

RESPONSE: The Authors added information in the Materials and Methods section about the number of repetitions performed in the WST-1 test.

Cytotoxicity of meloxicam on melanoma cells was assessed using the Cell Proliferation Reagent I (WST-1). Cells were seeded in 96-well plates (2.5 × 10³ cells per well) and incubated in culture medium for 24h. After incubation, the culture medium was removed, and meloxicam solutions in medium (100 µL per well) were added for 24h, 48h or 72h. Three hours before measurement, WST-1 reagent (10 µL per well) was added into each well. Absorbance was recorded at 440 nm and 650 nm using an Infinite 200 PRO microplate reader (TECAN, Männedorf, Switzerland). This experiment was performed in three independent repetitions. The results were expressed as a percentage relative to the control.

COMMENT: Page 2, Line 86: Correct the misspelled word “synthesise” to "synthesize."

RESPONSE: The Authors have corrected a misspelling of the word synthesize as suggested by the Reviewer: Tests were performed on cell types differentiated in their ability to synthesize the melanin biopolimer – amelanotic C32 and melanotic COLO 829.

COMMENT: On what basis was the meloxicam dosage concentration interval chosen in cell viability tests? Was it based on the design of previously published studies or on laboratory experience? Please indicate clearly in the text.

RESPONSE: The concentration range of meloxicam tested was selected on the basis of both previous laboratory experience based on conducting studies using normal skin cells with varying degrees of pigmentation and literature data describing the use of MLX in prostate cancer therapy, which used an analogous concentration range and time interval.

Prior to analysis, melanoma cells were incubated for 24, 48, and 72h with meloxicam in wide concentration range – 10 – 800 µM, selected on the basis of previously conducted studies on normal skin cells with varying degrees of pigmentation as well as literature data using MLX in the treatment of other cancers in an analogous concentration range and time interval [x,y].

COMMENT: In the title of Figure 2, the explanation regarding Figure 2C is completely missing. Please add it to the Figure caption. In addition, Figure 2C is quite blurred. Please increase the figure resolution if possible.

RESPONSE: The Authors added a missing description of Figure 2c.

Figure 2. The impact of meloxicam (MLX) on amelanotic (C32) (a), and melanotic (COLO 829) (b) melanoma cell cycle. All analyses were performed after a 48h incubation of the cells with MLX at
a concentration of 300, 500 or 700 µM. Representative histograms from the analysis (c) depict the distribution of the percentage of cells in the different phases of the cell cycle. Bar graphs show the mean value ± SD of three independent experments; * p < 0.05; ** p < 0.01.

COMMENT:. Page 4, Line 134 and Page 5, Line 143: Is it Figures 3A and 3C rather than Figures 2A and 2C?

RESPONSE: As the Authors have added Figure 1 there has been a change in the numbering order - these are now Figure 4 and 4a.

The obtained results imaged of Figure 4a revealed that incubation with MLX increased oxidative transitions in amelanotic and melanotic melanoma cells, however the observed effect was stronger in amelanotic cell line.

As greater disruption of intracellular oxido-reductive homeostasis has been reported for amelanotic melanoma, the content of oxygen free radicals (ROS) in this cell line was sequentially investigated using the fluorimetric spectrometry technique and the H2DCFDA probe (Figure 4c).

COMMENT: The histograms within Figures 3B and 4B are very difficult to read. Please increase the resolution of these Figures.

RESPONSE: The Authors inform that the images have been reinserted and are in 300dpi quality. The problem is the conversion of the Word file to pdf, where the image quality degrades significantly. However, it is possible to view the original images, which are of good quality, in the journal.

COMMENT: Please indicate clearly what is presented in Figure 4A. The caption of Figure 4 is a little bit confusing because the letter “a” and its explanation are missing.

RESPONSE: The Authors have added “a” description of the element a on Figure 4 that now is Figure 5.

Figure 5. The effect of meloxicam (MLX) on mitochondrial transmembrane potential in amelanotic C32, and melanotic, COLO 829 melanoma (a). All analyses were performed after a 48h incubation of the cells with MLX at a concentration of 300, 500 or 700 µM. Bar graphs show the mean value ± SD of three independent experments; ** p < 0.01. The scatterplots obtained from the analysis (b) present the populations of cells with depolarized (Q1lr) and polarized (Q1ur) mitochondria.

COMMENT: Please change the title of Subsection 2.5 to sound more general. For example: “Assessment of meloxicam-induced apoptosis in melanoma cells”.

RESPONSE: The Authors have changed the title of the 2.5 subsection as suggested by the Reviewer.

COMMENT: Page 6, Line 178: Is it Figure 5A rather than Figure 4A?

RESPONSE: According to the newly applicable numbering, this is Figure 6A, which has been corrected in the text.

COMMENT:. The Scatter plots within Figure 5B are very difficult to read and understand. Please increase the resolution significantly to enable their interpretation.

RESPONSE: The Authors inform that the images have been reinserted and are in 300dpi quality. The problem is the conversion of the Word file to pdf, where the image quality degrades significantly. However, it is possible to view the original images, which are of good quality, in the journal.

COMMENT:. Page 7, Line 193: Is it Figure 6 rather than Figure 5?

RESPONSE: According to the newly applicable numbering, this is Figure 7, which has been corrected in the manuscript.

COMMENT: At the end of the section Conclusions, please clearly indicate the limitations of the present research and future research directions regarding the antitumor potential of meloxicam in the treatment of melanoma and other malignancies.

RESPONSE: The Authors modified the Conclusions section as suggested by the Reviewer.

In conclusion, MLX demonstrates significant potential as an adjunct in anticancer therapy due to its multifaceted mechanisms of action. By inhibiting tumor cell proliferation, migration, and invasion, and promoting apoptosis through both COX-2-dependent and independent pathways, meloxicam offers a promising approach to targeting cancer progression. Its ability to enhance immunotherapy responses and disrupt hypoxia-related survival pathways further underscores its therapeutic value. However, the present research is limited by a scarcity of clinical trials, particularly in human patients, and a reliance on in vitro and animal models, which may not fully replicate the complexity of human malignancies. Moreover, the precise molecular mechanisms underlying meloxicam’s antitumor effects remain incompletely understood. Future research should focus on well-designed clinical studies to assess the efficacy and safety of meloxicam in cancer patients, as well as investigations into its mechanisms of action, optimal dosing strategies, and potential synergistic effects with other anticancer agents. Expanding research into its use in veterinary oncology, especially in canine cancers such as melanoma, may also offer valuable insights and translational relevance.

Round 2

Reviewer 3 Report

Comments and Suggestions for Authors

The authors have successfully implemented all the requested changes, so I am pleased to recommend the publication.